# Amplified fluorogenic immunoassay for early diagnosis and monitoring of Alzheimer's disease from tear fluid

Sojeong Lee [1], Eunjung Kim[2,3], Chae-Eun Moon [4], Chaewon Park[1], Jong-Woo Lim[1], Minseok Baek [5], Moo-Kwang Shin[1], Jisun Ki[1], Hanna Cho[6] ✉, Yong Woo Ji [4] ✉ & Seungjoo Haam [1] ✉

Accurate diagnosis of Alzheimer's disease (AD) in its earliest stage can prevent the disease and delay the symptoms. Therefore, more sensitive, non-invasive, and simple screening tools are required for the early diagnosis and monitoring of AD. Here, we design a self-assembled nanoparticle-mediated amplified fluorogenic immunoassay (SNAFIA) consisting of magnetic and fluorophore-loaded polymeric nanoparticles. Using a discovery cohort of 21 subjects, proteomic analysis identifies adenylyl cyclase-associated protein 1 (CAP1) as a potential tear biomarker. The SNAFIA demonstrates a low detection limit (236 aM), good reliability ($R^2 = 0.991$), and a wide analytical range (0.320–1000 fM) for CAP1 in tear fluid. Crucially, in the verification phase with 39 subjects, SNAFIA discriminates AD patients from healthy controls with 90% sensitivity and 100% specificity in under an hour. Utilizing tear fluid as a liquid biopsy, SNAFIA could potentially aid in long-term care planning, improve clinical trial efficiency, and accelerate therapeutic development for AD.

Alzheimer's disease (AD) is the most common neurodegenerative disease, but its pathogenesis remains unclear[1,2]. In the absence of a cure, the best treatment for AD is medication to slow the progression of symptoms and preserve as much of the patient's cognitive function as possible. Early diagnosis is critical to timely medication and improved prognosis, as starting treatment early in the course of the disease can maximize the effectiveness of preserving the patient's residual function and slowing the onset and progression of dementia symptoms[3,4]. Diagnosis of AD is based on observation of clinical signs and symptoms, testing of neurocognitive function, and examination of changes in AD-promoting biomarkers that may reflect the early stages[5–8]. Well-established methods include positron emission tomography (PET)-based brain function imaging and cerebrospinal fluid (CSF) analysis, which detects deposition of AD biomarkers such as amyloid-β and tau proteins in the brain[9,10]. However, PET scans are time-consuming and expensive, and the number of specialized healthcare facilities with access to PET scanners is limited. CSF testing requires samples to be collected via lumbar puncture, which is a painful and uncomfortable procedure for patients. These techniques are particularly limited as first-line AD diagnosis tests because they require invasive interventions, such as fluoridated tracer injections, exposure to radiation, and lumbar puncture, which can have adverse side effects[11]. Therefore, the development of biosensors that can sensitively and selectively detect AD biomarkers in a non-invasive manner while rapidly tracking changes in biomarker concentrations over time can improve clinical confidence in diagnosing from pre-symptomatic and prodromal stages to dementia.

[1]Department of Chemical and Biomolecular Engineering, Yonsei University, Seoul 03722, Republic of Korea. [2]Division of Bioengineering, Incheon National University, Incheon 22012, Republic of Korea. [3]Department of Bioengineering & Nano-bioengineering, Research Center for Bio Materials and Process Development, Incheon National University, Incheon 22012, Republic of Korea. [4]Department of Ophthalmology, Yongin Severance Hospital, Yonsei University College of Medicine, Yongin 16995, Republic of Korea. [5]Department of Neurology, Wonju Severance Christian Hospital, Yonsei University Wonju College of Medicine, Wonju 26426, Republic of Korea. [6]Department of Neurology, Gangnam Severance Hospital, Yonsei University College of Medicine, Seoul 06273, Republic of Korea. ✉e-mail: IGUHANNA@yuhs.ac; LUSITA30@yuhs.ac; haam@yonsei.ac.kr

Immunoassay is a technique that allows the quantitative analysis and detection of specific molecules (e.g., proteins, pathogenic antigens, antibodies, and small molecules) with high accuracy and sensitivity. It is one of the most widely used diagnostic techniques because it offers the potential for high throughput screening of diseases[12]. In order to overcome the technical challenges of shortening analysis time, improving bioanalytical performance, and increasing reproducibility, numerous approaches have been attempted by integration with functionalized nanomaterials and nanopatterns[13,14]. For example, lateral flow assays[15–18] using nanozymes as signal catalysts have shown enhanced sensitivity and wide dynamic detection range compared to conventional gold nanoparticle-based assays. In addition, various attempts have been made to introduce nanomaterials into sandwich immunoassays to achieve early diagnosis of AD in clinical samples, including electrical sensors using a densely aligned carbon nanotube pattern[19], a disposable microfluidic platform based on immunomagnetic capture system[20], and a surface plasmon resonance imaging sensor modified with antibody-mimetic peptoid nanosheets[21]. Despite the efforts to develop high-performance diagnostics applicable to these clinical samples, the limitations of AD diagnosis using invasive specimens such as CSF and blood still exist. Therefore, there is a desperate need to develop an easily accessible and sensitive sensing system that can distinguish the progressive stages of AD with high accuracy and cost-effectiveness using non-invasive biopsies directly related to disease monitoring.

Body fluids, such as saliva[22,23], sweat[24,25], urine[26], and tears, were emerging as non-invasive alternatives for clinical self-diagnosis and routine testing[27]. Among them, tear fluid is known to be closely associated with certain neurological disorders, including AD, Parkinson's disease, and multiple sclerosis. Tears are rich in soluble biomolecules, including proteins, peptides, metabolites, and nucleic acids[28,29] and contain very little albumin, which is typically found in large quantities in the blood. Thus, they do not require protein filtration or centrifugation procedures and are less prone to cross-reactivity issues due to excess proteins than blood or plasma. Unlike blood or CSF collection, tear fluid can be collected non-invasively, making it a cost-effective and relatively easy fluid to administer without the assistance of a healthcare professional. In addition, significant correlations have been reported between the expression levels of specific protein biomarkers (e.g., amyloid-β and tau proteins) in tears and the stage of AD progression[30–33]. These studies suggest that tear fluid is a suitable liquid biopsy for predicting and monitoring AD progression. However, the amount of tear fluid obtained from the eyes in a single instance is very small[32,34], and the protein concentration is also lower than that of serum. Therefore, the application of selective concentration and separation systems for target biomolecules enables the detection of target biomarkers in tear fluids, even under small volume and low concentration conditions.

Herein, we propose a diagnostic tool, a self-assembled nanoparticle-mediated amplified fluorogenic immunoassay (SNAFIA), for the highly sensitive detection of biomarker candidates in AD clinical tear samples. In this study, a total of 60 participants are used and organized into two distinct cohorts: discovery and verification (Fig. 1). The discovery cohort consists of a total of 21 samples and identifies a potential biomarker, adenylyl cyclase-associated protein 1 (CAP1), through proteomic experiments. The verification cohort consists of 39 samples in total, and the CAP1 protein identified in the discovery phase is applied to the SNAFIA immunoassay proposed in this study. Specifically, the SNAFIA assay consists of antibody-immobilized magnetic nanoparticles (Ab-MNPs) and polymeric nanoprobes (Ab-PNPs), which form sandwich-structured immunocomplexes (Ab-MNP-CAP1-Ab-PNP). SNAFIA can reliably detect CAP1 protein down to the attomolar concentration level and quantitatively measure the concentration of the AD biomarker candidate in tear fluid within 1 h. Finally, SNAFIA can differentiate clinically diagnosed AD patients from normal controls by directly measuring the expression level of CAP1 protein in tear fluid through enzyme-free fluorescence signal amplification and a simple test workflow. Therefore, this SNAFIA system could provide an excellent alternative to existing commercial immunoassay kits and blood diagnostic tests for diagnosing AD, and could potentially be applied to provide disease monitoring systems utilizing other body fluids.

## Results and discussion

### Discovery of human tear fluid biomarker by proteomic analysis

The present study was methodically structured with a two-pronged approach, encompassing both the discovery and verification stages. The discovery phase was tailored to pinpoint potential biomarkers, while the verification phase affirmed the expression levels of potential biomarkers using our proposed immunoassay. We utilized two distinct cohorts of human tear samples for the proteomic experiments: the discovery cohort and the verification cohort. All subjects received [18]F-florbetaben PET imaging for amyloid-β measurement and were conducted Mini-Mental State Examination (MMSE). Specifically, the discovery cohort was comprised of tear samples from 7 healthy controls (HC), 7 mild cognitive impairment (MCI), and 7 AD participants, totaling 21 samples (Fig. 2a and Supplementary Table 1).

To investigate novel possible tear biomarkers for AD diagnosis and progression monitoring, we performed a high-resolution and comprehensive proteomic analysis using tear samples of the discovery cohort (Fig. 2a). Using in-depth proteome profiling, we identified 75 differentially expressed proteins (DEPs), consisting of 64 progressively up-regulated and 11 gradually down-regulated proteins, in the tears of individuals with MCI and AD compared to HC. 'Progressively up-regulated' means an increasing trend in fold change from the MCI/HC comparison to the AD/HC comparison. This means that the protein's expression level increased more prominently when comparing AD patients to HC than when comparing MCI patients to HC. 'Gradually down–regulated' means a decreasing trend in fold change from the MCI/HC comparison to the AD/HC comparison. This indicates that the protein's expression level decreased more significantly when comparing AD patients to HC than when comparing MCI patients to HC (Fig. 2b and Supplementary Table 2). Gene ontology-based biological process analysis revealed that these proteins were associated with immune response, regulation of mRNA metabolism, and metabolic processes. Among them, we selected the CAP1 protein, whose relative protein expression was significantly changed in tears from both MCI and AD patients (1.72 and 1.86 compared to HC, respectively) (Fig. 2c).

In the context of oncology, CAP1 has been implicated in tumor progression and metastasis across various cancer types[35]. Additionally, it is known that its presence in vascular and macrophage membranes influences the inflammatory responses of monocytes[36], potentially contributing to conditions like coronary artery disease, immune disorders, metabolic disturbances, and pulmonary diseases. In particular, CAP1 is pivotal for neuronal actin dynamics and the proper functioning of growth cones[37], any perturbation in CAP1 levels or function might contribute to synaptic dysfunction, a characteristic feature of AD. Furthermore, a recent study highlighted differential expression of CAP1 in exosomes derived from the serum of AD patients[38]. Although the precise role of CAP1 in AD remains to be elucidated, its altered expression patterns hint at a potential involvement in the disease's pathogenesis or progression. Meanwhile, there are a few reports on the detection of CAP1 in body fluids such as serum[38,39] and bronchoalveolar lavage[40]. Given the critical role of CAP1 in the pathogenesis of AD[37,38,41], the sensitive detection of CAP1 protein in tears could provide a valuable alternative to existing diagnostic techniques for AD and aid in the development of new therapeutic interventions.

In Fig. 2b, c, while our study primarily focused on the protein CAP1, several proteins with roles in defense mechanisms and the

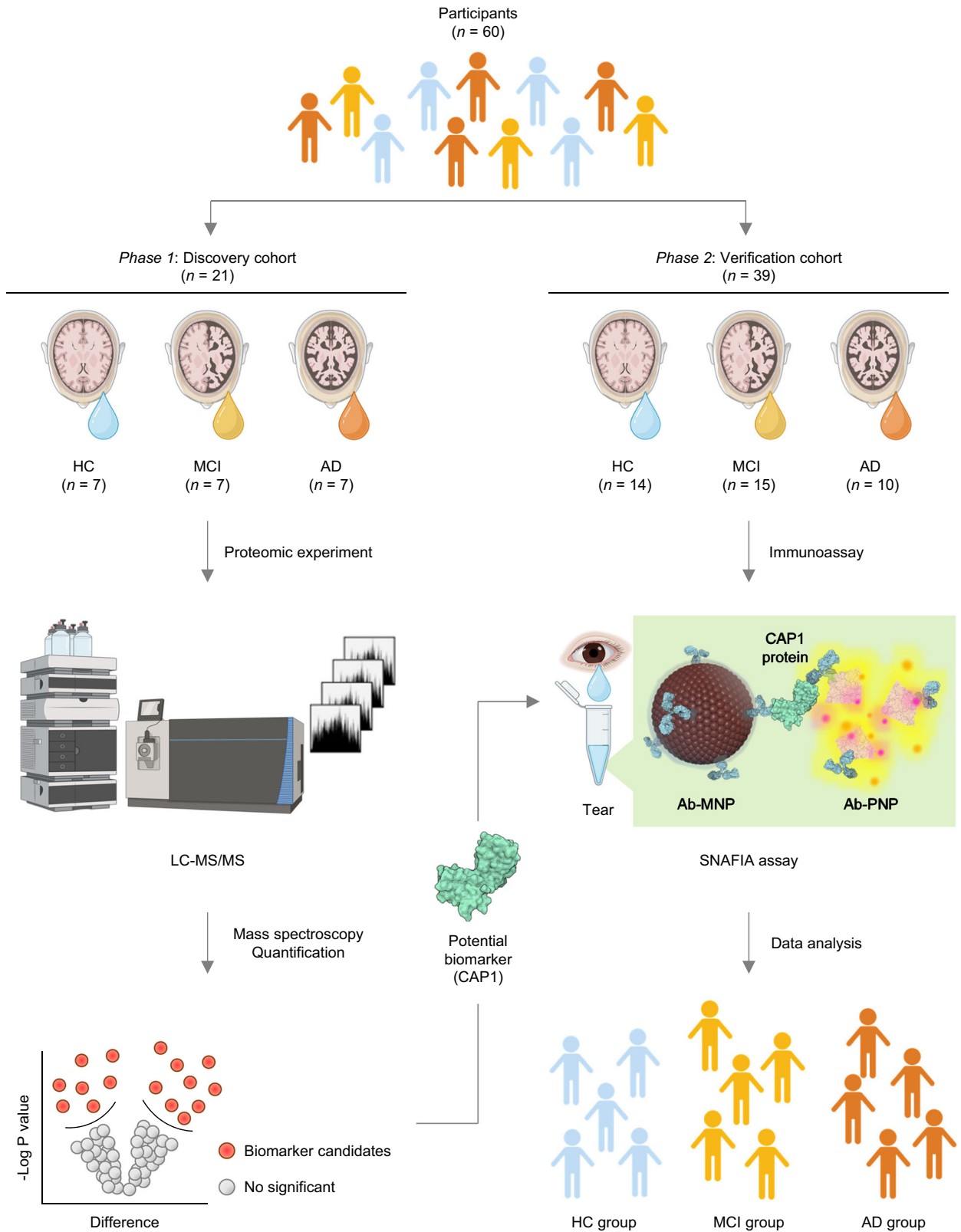

chemical barrier are also depicted. For instance, proteins like LCP1 (Lymphocyte Cytosolic Protein 1), CORO1A (Coronin 1A), HMGB1 (High Mobility Group Box 1), RETN (Resistin), NAMPT (Nicotinamide Phosphoribosyl transferase), and HSPA5, HSPB1 (Heat Shock Proteins) have known roles in immune responses, while others like SCGB1D1 and SCGB2A1 (Secretoglobins) possess immunomodulatory properties. Additionally, proteins such as LACRT (Lacritin) are specific to the eye

and influence tear composition and defense capabilities of tears. LCN1 (Lipocalin-1) might play a role in modulating the chemical environment of tears, indirectly contributing to its defense mechanisms. Recent literature has indeed highlighted changes in the components of the chemical barrier and the network of antimicrobial and immunomodulatory peptides in tear fluid in the context of AD[30]. The altered composition of the chemical barrier, along with the reduced level of

**Fig. 1 | Workflow of a discovery and verification cohort study for early diagnosis and monitoring of Alzheimer's disease.** Sixty human tear fluid samples were utilized in two cohorts: a discovery cohort and a verification cohort. The discovery cohort was comprised of tear samples from 7 healthy controls (HC), 7 patients with mild cognitive impairment (MCI), and 7 patients with Alzheimer's disease (AD) for a total of 21 tear samples. Proteomic analysis using mass spectrometry identified adenylyl cyclase-associated protein 1 (CAP1) as a promising potential biomarker. The verification cohort consisted of 14 HC, 15 MCI patients, and 10 AD patients, with a total of 39 tear samples. A self-assembled nanoparticle-mediated fluorescence immunoassay (SNAFIA) was applied to detect the target protein, CAP1, in human tear fluid samples. The presence of the CAP1 protein generated a sandwich immunocomplex with an antibody-conjugated magnetic nanoparticle (Ab-MNP) and an antibody-conjugated polymeric nanoprobe (Ab-PNP), enabling the discrimination of the three groups of participants based on the analysis of the fluorescence signals from the SNAFIA assay. The schematic was created with BioRender.com.

defense proteins, might imply an increased risk of ocular infections in AD patients. However, there hasn't been a reported increase in ocular infections in AD patients in the scientific literature.

Overall, it is noteworthy that while the expression levels of CAP1 did not exhibit the most pronounced changes in comparison to other proteins, they did manifest a discernible and consistent trend of incremental expression from HC individuals to those with MCI, and subsequently to AD patients. Such a definitive increasing pattern across disease progression is crucial for early detection rather than a mere high expression at a particular disease stage. The stepwise elevation in CAP1 levels underscores its potential utility as a diagnostic marker, particularly for biosensing platforms designed for the identification of diseases in their early stages. The distinctive expression patterns of CAP1 observed in the tear fluid of MCI and AD patients prompted us to plan further experiments including a verification cohort.

## Design a SNAFIA platform comprising functionalized nanoparticles

For the highly sensitive detection of target protein, we designed an immunoassay utilizing two functionalized nanoparticles: magnetic nanoparticles (MNPs) and polymeric nanoprobes (PNPs) (Fig. 3a). MNPs have been extensively used in immunoassays because their magnetic properties, which can be tuned by an external magnetic field, provide a useful method to quickly isolate target molecules from complex biological solutions. To confirm the existence of antibodies on the MNPs surface, the immobilized antibodies were labeled with immunogold (Ab-AuNPs) (Fig. 3b). In Fig. 3c, the transmission electron microscopy (TEM) image shows Ab-AuNPs localized on the periphery of Ab-MNPs, indicating the successful conjugation of antibodies to the MNPs. The absence of Ab-AuNPs in the background also implies that the AuNPs were well removed by magnetic separation. Elemental maps obtained using TEM coupled with energy-dispersive X-ray spectroscopy further confirmed that iron and oxygen cores comprise the Ab-MNPs along with silica shells, and that gold is localized on the surface of Ab-MNPs (Supplementary Fig. 2).

$Fe_3O_4$ nanoparticles are promising nanomaterials for separating non-magnetic components due to their high magnetic susceptibility, which measures the ability of a material to be magnetized in the presence of a magnetic field. To evaluate this property, we obtained magnetic hysteresis curves of MNPs series using vibrating sample magnetometry (VSM) at 25 °C (Supplementary Fig. 3 and Supplementary Table 4). As each layer was incorporated into the MNPs surface, the observed saturation magnetization ($M_s$) values decreased by about 27–42% compared to the intact MNPs, which was attributed to the non-magnetic silica/PEG polymer and biomolecular coating. Nevertheless, the synthesized Ab-MNPs were found to still retain their superparamagnetic behavior (Fig. 3d). This indicates that they exhibit durable magnetic responsiveness under an applied magnetic field, allowing them to be used to separate target molecules from the reaction mixtures using an external magnet.

For improved sensitivity and selective fluorescence signal amplification, PNPs containing Förster resonance energy transfer (FRET) dye pairs on their hydrophobic membranes were applied to the SNAFIA assay (Fig. 3e). This isolates the FRET dyes within the membrane, which serves as a physical barrier from external environmental and degradation factors, improving the background-to-signal ratio and enabling consistent and reliable results. To generate FRET, 3,3′-dioctadecyloxacarbocyanine perchlorate (DiO) as a donor and 1,1′-dioctadecyl-3,3,3′,3′-tetramethylindocarbocyanine perchlorate (DiI) as an acceptor were encapsulated into the membrane of Ab-PNPs (Fig. 3f). Thus, the SNAFIA detection system is based on a signal shift from high to low FRET due to the dissociation of DiO and DiI fluorophores, triggered by lysis buffer. As shown in Fig. 3j, upon excitation at 475 nm, fluorescence emission at 565 nm was observed for Ab-PNPs, whereas the treatment of Triton X-100 as a lysis buffer induced a sudden increase in fluorescence emission at 500 nm.

To maximize the FRET signal change, we varied the dye loading amount (4–100 μg), loading molar ratio of DiO and DiI (5:1, 2:1, 1:1, 1:2, and 1:5), and detection time (0–60 min) to find the optimal conditions (Supplementary Figs. 6–8). As the amount of loading dye increased, the size of Ab-PNPs gradually increased, and their entrapment efficiency was in the range of 50–70%. A loading molar ratio of 1:1 of the two dyes resulted in a good size distribution of Ab-PNPs and the largest FRET signal change compared to other conditions. The FRET signal change of Ab-PNPs also reached a saturation point about 15 min after treatment with lysis buffer. Therefore, based on these control experiments, a standard protocol for the SNAFIA test was established, which included 10 μg of DiO and DiI loading dyes at a 1:1 molar ratio and 15 min incubation time. This ensures rapid and complete destruction of Ab-PNPs and produces the maximum signal change. Finally, we checked the signal amplification effect in different pH buffers (MES, pH 5.3; PBS, pH 7.4; Tris-HCl, pH 8.1), deionized water (DW), and artificial tear fluid (ATF), exhibiting similar signal changes regardless of buffer conditions (Supplementary Fig. 9). These results suggest that the signal amplification via the FRET effect of Ab-PNPs can be applied to various reaction buffers and body fluids.

## Assessment of the SNAFIA using potential biomarkers in biofluid

We evaluated the detection performance of SNAFIA using CAP1 protein, which is considered to be a potential biomarker for AD. To do this, Ab-MNPs and Ab-PNPs were first coated with CAP1-recognizing IgG antibodies to selectively capture the target protein. Ab-MNPs (0.1 mg/mL) and Ab-PNPs (0.1 mg/mL) were added to various concentrations of CAP1-containing PBS solution (0.0128–5000 fM) and incubated at 37 °C for 30 min to induce an immunocomplex formation. After washing with PBS three times, the residual immunocomplexes were treated with 1% (v/v) of Tx-100 and incubated for 15 min to solubilize Ab-PNPs. The fluorescence emission signals of the FRET dyes derived from the lysed Ab-PNPs were recorded at 500 nm (excitation at 475 nm) using a laboratory plate reader (Fig. 4a). The SNAFIA test exhibits linearity over five orders of magnitude from 64.0 fM to 200 pM of the target protein ($R^2 = 0.975$), reflecting the ability to quantify a wide range of concentrations (black circles in Fig. 4b and Supplementary Table 5). Linear regression analysis showed the limit of detection (LOD) based on the three-sigma (3σ) method and the limit of quantification (LOQ) based on the ten-sigma (10σ) method of the SNAFIA test for CAP1 were 0.283 and 0.447 fM, respectively (Supplementary Table 5).

To confirm the feasibility of testing the human tear fluid samples, we conducted the SNAFIA test on CAP1 protein spiked in artificial tear fluid (ATF). Before the concentration-specific detection evaluation, CAP1 protein was prepared at a concentration of 1 nM in PBS and ATF

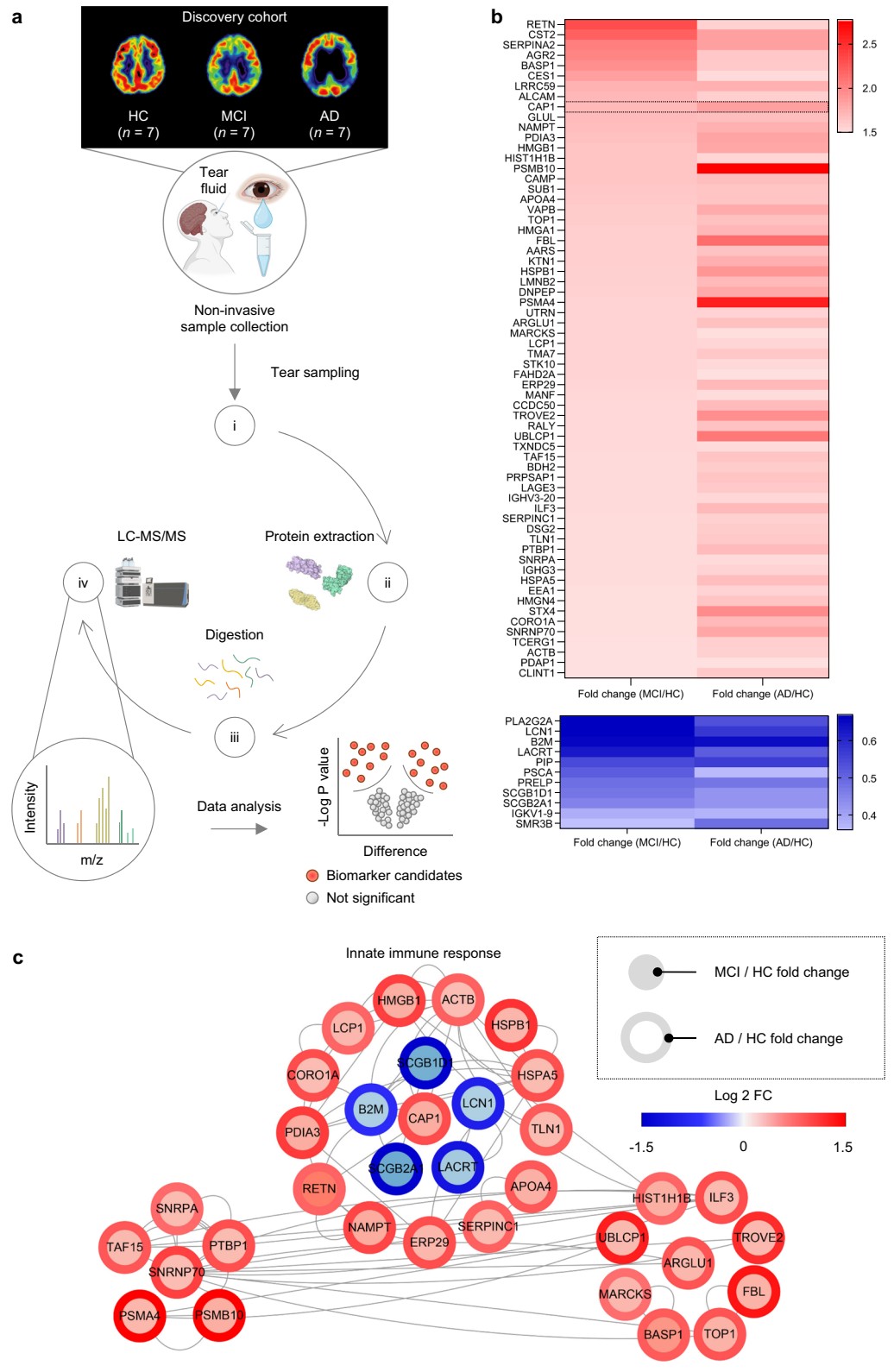

(Supplementary Fig. 10a). The fluorescence intensity of SNAFIA treated with ATF showed a 96% high similarity performance from the results in PBS (Supplementary Table 6) and the LODs calculated via linear regression ($R^2 = 0.944$) showed similar values of 0.282 and 0.236 fM, respectively (Supplementary Table 5). Thus, SNAFIA has been prepared to detect CAP1 protein as low as 0.3 fM utilizing human tear fluid samples.

Next, we compared the detection performance of the SNAFIA assay with a classical immunosensing platform, enzyme-linked immunosorbent assay (ELISA), using the same antibody and target protein. In the ELISA test, the immunocomplex is composed of a capture antibody immobilized on the bottom of the microplate and a detection antibody is conjugated to an enzyme that catalyzes a chromogenic substrate to generate a color change. As shown in Fig. 4c and

**Fig. 2 | Discovery of human tear fluid biomarkers for AD by proteomic analysis.** **a** Schematic of the experimental design for global proteome profiling by tandem mass tag labeling using human tear fluid samples from the discovery cohort. (i) Tear fluids were collected non-invasively using a polyester wick. (ii) Proteins of the pooled tear fluids digested into peptides by in-solution digestion. (iii) Each peptide sample was labeled with tandem mass tags. Peptide separation was carried out using high-pH reverse-phase liquid chromatography fractionation. (iv) For the proteome profiling analysis, fractionated peptides were analyzed using the Q Exactive orbitrap hybrid mass spectrometer. Full mass spectrometry data were acquired using the Proteome Discoverer software version 2.1. **b** Heatmap of differentially expressed proteins (DEPs) determined by comparing the relative expression levels in patients with MCI or AD to the HC group (MCI/HC or AD/HC).

Proteins exhibiting fold changes greater than 1.5 classified as upregulated (top) and proteins showing a fold change less than 0.67 classified as downregulated (bottom). Statistical analysis was performed for each protein using a $t$-test ($p < 0.01$). **c** Protein–protein interaction network with significantly enriched biological processes generated from proteins differentially expressed in tear fluid from patients with MCI and AD compared with HC. The inner and outer gray circles represent the relative protein expression (fold change) of each AD and MCI group compared to the HC group, with values close to 1.5 and −1.5 on the log2 fold change scale shown in red and blue, respectively. The colors of the nodes represent proteins that were significantly increased (red) or decreased (blue) in MCI or AD. Gray lines between nodes indicate biological or physical interactions between proteins. Schematics were created with BioRender.com.

Supplementary Fig. 11, we obtained increased absorbance signals with increasing concentrations of CAP1 (0–25000 nM), yielding a narrower dynamic range compared to SNAFIA. As such, SNAFIA demonstrated approximately $10^2$-fold improved detection range and $10^7$-fold reduced LOD value compared to ELISA. Furthermore, we designed a half-SNAFIA (hSNAFIA) platform where Ab-PNPs were replaced with AlexaFluor®488 (AF488)-conjugated antibodies to compare the effectiveness of signal amplification using Ab-PNPs (Supplementary Fig. 12). Immunocomplexes formed by the same protocol as SNAFIA were incubated with a human CAP1 antibody followed by an AF488-conjugated secondary antibody, and excess antibodies were removed by applying a magnet. Target concentration-responsive fluorescence signals were then obtained in the hSNAFIA test with varying concentrations of CAP1 (0.256 fM to 100 pM). The results showed that hSNAFIA exhibited a dynamic range in the concentration range intermediate between SNAFIA and ELISA. As summarized in Supplementary Table 7, the SNAFIA method showed a 10 and $10^7$ times improved detection limit compared to hSNAFIA and ELISA, respectively, indicating that encapsulation of a large amount of FRET dyes within the hydrophobic membrane of Ab-PNPs and the selective release of dyes by Triton X-100 allowed for the generation of enhanced signals.

To provide robustness against non-specific responses, we investigated the selectivity of the SNAFIA-based detection method using AD-related proteins of CAP1, apolipoprotein E (APOE), and acetylcholine esterase (ACHE). APOE and ACHE proteins, along with CAP1, have been involved in the pathogenesis of AD and studied as therapeutic strategies. As shown in Fig. 4d, non-target proteins present at a concentration greater than 10-fold in PBS buffer had no significant effect on SNAFIA signal generation, resulting in a fluorescence signal that was selectively amplified for the CAP1 target protein. This demonstrates that the exhibits distinct target detection performance within specimens containing a mixture of both target (CAP1 protein) and non-target proteins (APOE and ACHE proteins), with a $p$-value less than 0.0001.

As depicted in Supplementary Fig. 13a, to establish the versatility of the SNAFIA assay for a variety of human biofluids, including tear fluid and serum, we conducted spiking experiments involving the introduction of APOE protein into human serum. APOE is one of the components of lipoproteins that transport and metabolize lipids, and polymorphisms in the APOE gene, especially the APOE e4 allele, are known to profoundly affect the risk of sporadic AD[41]. Its association with an elevated risk of AD development is well-documented in numerous studies[1,42–44]. To evaluate the utility of SNAFIA in human serum, Ab-MNPs and Ab-PNPs were coated with IgG antibodies that recognize all APOE proteins, not just the APOE4 subtype. Before the tests, serum solutions were prepared at 0, 10, 25, 50, and 100% (v/v) by dilution in PBS, and APOE was added to each serum solution at a concentration of 1 nM (Supplementary Fig. 14a). We observed a gradual reduction in fluorescence intensity as the amount of serum increased, but the signal in the 10-fold diluted serum was similar to that in PBS. Based on these results, we compared the sensitivity of the

SNAFIA assay by spiking APOE protein into 10% (v/v) human serum solution (Supplementary Fig. 14b). Next, obtained samples by spiking APOE protein at concentrations ranging from 12.8 aM to 5 pM into PBS and 10% (v/v) human serum solution (Supplementary Fig. 13b). The LODs calculated via linear regression showed similar values of 0.0542 and 0.0586 fM, respectively (Supplementary Table 8). Thus, SNAFIA has the ability to directly detect APOE at concentrations as low as 0.06 fM, confirming its applicability to human-derived biological samples. We also compared the sensitivity of SNAFIA to hSNAFIA and ELISA using the APOE protein. As shown in Supplementary Figs. 13c and 15, we obtained increased absorbance signals with increasing concentrations of APOE (0–2000 nM), yielding a narrower dynamic range compared to SNAFIA. Also, the SNAFIA method provided the best LOD and widest detection range compared to hSNAFIA and ELISA (10-fold and $10^7$-fold enhancement in LOD), similar to the improvement trend measured using CAP1 protein (Supplementary Table 8). As summarized in Supplementary Table 9, the SNAFIA method showed a 10 and $10^7$ times improved detection limit compared to hSNAFIA and ELISA, respectively.

The primary aim of our study was to demonstrate the efficacy of the SNAFIA test, an immunodiagnostic platform designed for detecting protein biomarkers in tears. Therefore, we additionally conducted the spiking experiments for APOE protein into ATF and for CAP1 protein into 10% (v/v) human serum solution. Similar to sensitivity tests, positive dose-responsive fluorescence signal changes for CAP1 protein in 10% (v/v) human serum solution and for APOE protein in ATF were identified using SNAFIA (Supplementary Figs. 16 and 17 and Supplementary Tables 5 and 8). This demonstrates that the SNAFIA exhibits distinct target detection performance to various human-derived biological samples.

Lastly, we investigated the selectivity of the SNAFIA along with APOE, depicted in Supplementary Fig. 13d. The SNAFIA platform also discriminated the presence of the target protein (APOE protein) in samples mixed with non-target proteins (CAP1 and ACHE proteins), with a $p$-value less than 0.005. Furthermore, reliable results for detecting various targeted proteins can be obtained using a set of surface-tailored Ab-MNPs and Ab-PNPs and SNAFIA can accurately and precisely detect CAP1 potential biomarker in tear fluid at the attomolar concentration level.

Notably, our proposed system is designed to have a shorter detection time compared to conventional diagnostic methods for detecting protein biomarkers. For example, an ELISA kit with a microplate pre-coated with antibodies takes about two hours to obtain test results after several procedures, including incubation, blocking, and washing steps. However, the SNAFIA platform allows Ab-MNPs and Ab-PNPs to simultaneously interact with the biomarker to rapidly form an immunocomplex, and then the free proteins and Ab-PNPs can be washed away by magnetic separation within 5 min, separated from the immunocomplex with only the target protein. The fluorescence signal can also be induced immediately with a lysis buffer. To analyze the overall run time of the detection system, we analyzed the signal of the SNAFIA test as a function of incubation time and found that a distinct

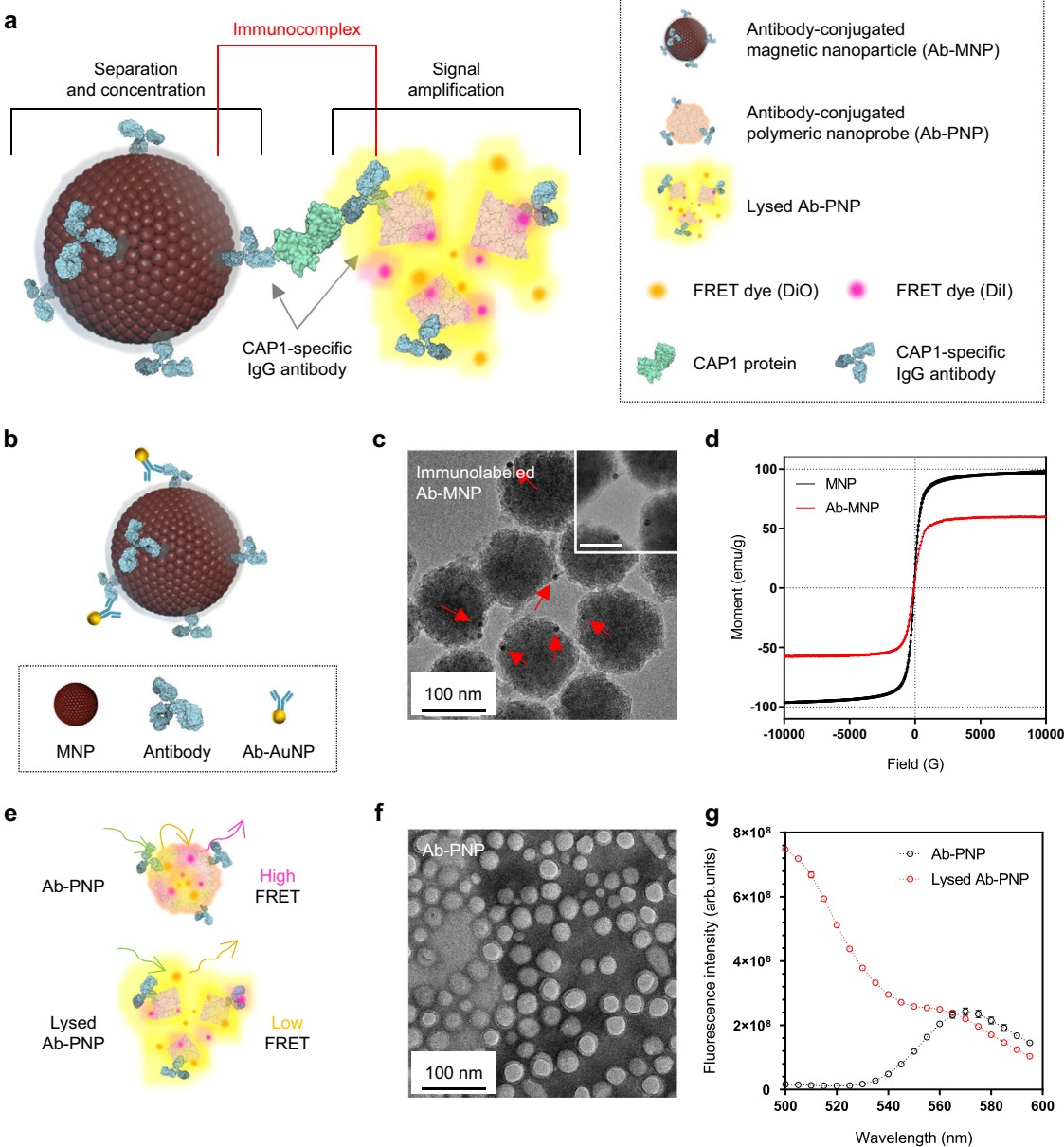

**Fig. 3 | Characterization of the synthesized Ab-MNPs and Ab-PNPs. a** Schematic of the immunocomplex constructed by Ab-MNPs and Ab-PNPs in the presence of the target protein (CAP1) in tear fluid. **b** Schematic representation of Ab-MNPs labeled with immunogold (Ab-AuNPs) for better visualization of the primary capture antibody bound to the MNPs. **c** Representative transmission electron microscopy (TEM) image of Ab-MNPs labeled with immunogold (Ab-AuNPs, approximately 10 nm in diameter). Red arrows indicate the localized AuNPs on the surface of Ab-MNPs. The scale bars indicate 100 nm and 50 nm (inset), respectively. **d** Magnetization curves of MNPs (black) and Ab-MNPs (red) obtained by vibrating sample magnetometer (VSM) at 25 °C. **e** Schematic of Förster resonance energy transfer (FRET) signal changes of Ab-PNPs induced by treatment with TX-100 surfactant as lysis buffer. **f** TEM image of Ab-PNPs negatively stained with 3% (w/v) phosphotungstic acid solution (pH 6.81). The scale bar represents 100 nm. **g** Emission fluorescence spectra of Ab-PNPs before (black circles) and after (red circles) treatment with lysis buffer (excitation at 475 nm). Data represent mean ± s.d. for three independent experiments. The representative images were taken from different samples and repeated at least 50 times independently collection with similar results. Schematics were created with BioRender.com.

signal occurs after 30 min, with a gradual increase in fluorescence intensity thereafter (Supplementary Fig. 18). Thus, SNAFIA provides test results within 1 h, including 30–35 min for biomarker-specific immunocomplex formation and purification and 15 min for fluorescent signal production by PNP degradation (Supplementary Fig. 8c). Overall, the in-solution SNAFIA assay does not require enzyme-catalyzed reactions, utilize the high surface area-to-volume ratio of nanomaterials to increase antigen-antibody binding efficiency, and relies on magnetic separation and a fluorescent dye release system. These features improve sensitivity and selectivity and simplify the entire detection steps while reducing the detection time to less than 1 h.

## Verification of diagnostic application of SNAFIA

Finally, we validated the clinical applicability of SNAFIA using tear fluids from a distinct verification cohort of 39 individuals. This cohort included 14 HC, 15 patients with MCI (known as preclinical AD), and 10 patients with definite AD (Fig. 1). All subjects received $^{18}$F-florbetaben PET imaging for amyloid-β measurement and were conducted MMSE. Detailed demographic characteristics of the subjects (gender, age, and MMSE score) are listed in Supplementary Table 10. Tear samples from these patients were used to perform the SNAFIA test. We employed the same method for tear collection in both cohorts to maintain uniformity in sample collection across different stages of our study. For

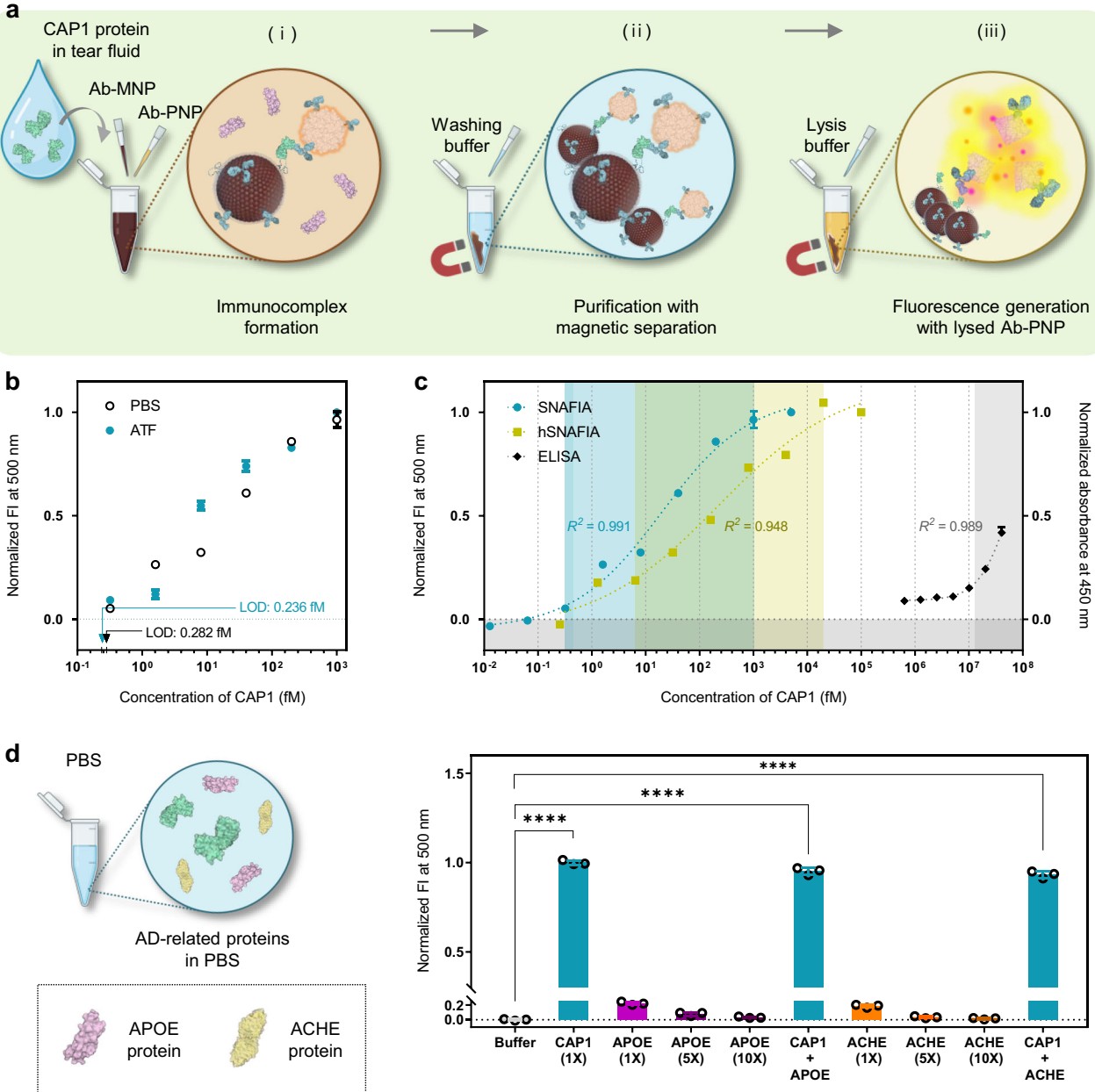

**Fig. 4 | Evaluation of the SNAFIA platform toward AD biomarker candidates in artificial tear fluid. a** Detailed process of SNAFIA test using tear fluid. First, (i) tears are combined with Ab-MNPs and Ab-PNPs to form a sandwich immunocomplex with a target protein. (ii) The immunocomplex is subsequently separated from the mixture using a magnet, removing non-target and excess Ab-PNPs. (iii) TX-100 surfactant is added as a lysis buffer to disrupt Ab-PNPs and allow to release of FRET dyes from the Ab-PNPs, amplifying the signal. **b** Normalized fluorescence intensity (FI) of SNAFIA with increasing concentrations of CAP1 protein spiked in phosphate-buffered saline (PBS, black circles) and artificial tear fluid (ATF, turquoise circles) solution (excitation at 475 nm, emission at 500 nm). The limit of detection (LOD) of SNAFIA in each condition was determined by three-sigma (3σ) and calculated to be 0.282 and 0.236 fM, respectively. **c** Fluorescence and absorbance signal responses of the SNAFIA (turquoise dots), half-SNAFIA (hSNAFIA, yellowish dots), and ELISA (black dots) tests with different concentrations of CAP1 protein ranging from $10^{-2}$ to $10^8$ fM. Normalized FI and absorbance data were fitted to the four-parameter logistic curve (dotted line), and the linear dynamic range of each test is shown by the shaded area. **d** Selectivity of SNAFIA for various AD biomarker candidates and their mixtures in PBS. The concentrations of CAP1 (turquoise bars), apolipoprotein E (APOE, magenta bars), and acetylcholinesterase (ACHE, orange bars) were varied to 1 pM (1×), 5 pM (5×), and 10 pM (10×). Statistical analysis was performed by multiple comparisons of Brown-Forsythe and Welch one-way analysis of variance tests (****$p < 0.0001$). The measurement was performed in triplicate, and all reported values represent mean ± s.d.; $n = 3$ repeated tests. Schematics were created with BioRender.com.

application to tear AD diagnosis, we built a SNAFIA system to detect the CAP1 biomarker candidate in tear fluid and analyzed the differences in fluorescence signals obtained from the three groups (Fig. 5b). As shown in Fig. 5c, the average intensity of the CAP1 detection signals in tear fluid samples from the MCI and AD groups was approximately 1.77 and 2.57 times higher than that of the HC group, respectively. We also performed receiver operator characteristic analysis to evaluate

the diagnostic power of SNAFIA and determine the optimal cutoff point to distinguish among HC, MCI, and AD groups (Fig. 5d). The area under the curve (AUC) values for the MCI and AD groups were 0.7619 and 0.9714, respectively, with $p$-values less than 0.0001. Thus, the SNAFIA test was able to accurately identify the MCI and AD patient groups from the HC participants with a sensitivity of 73.3% and specificity of 100%, and a sensitivity of 90% and specificity of 100%,

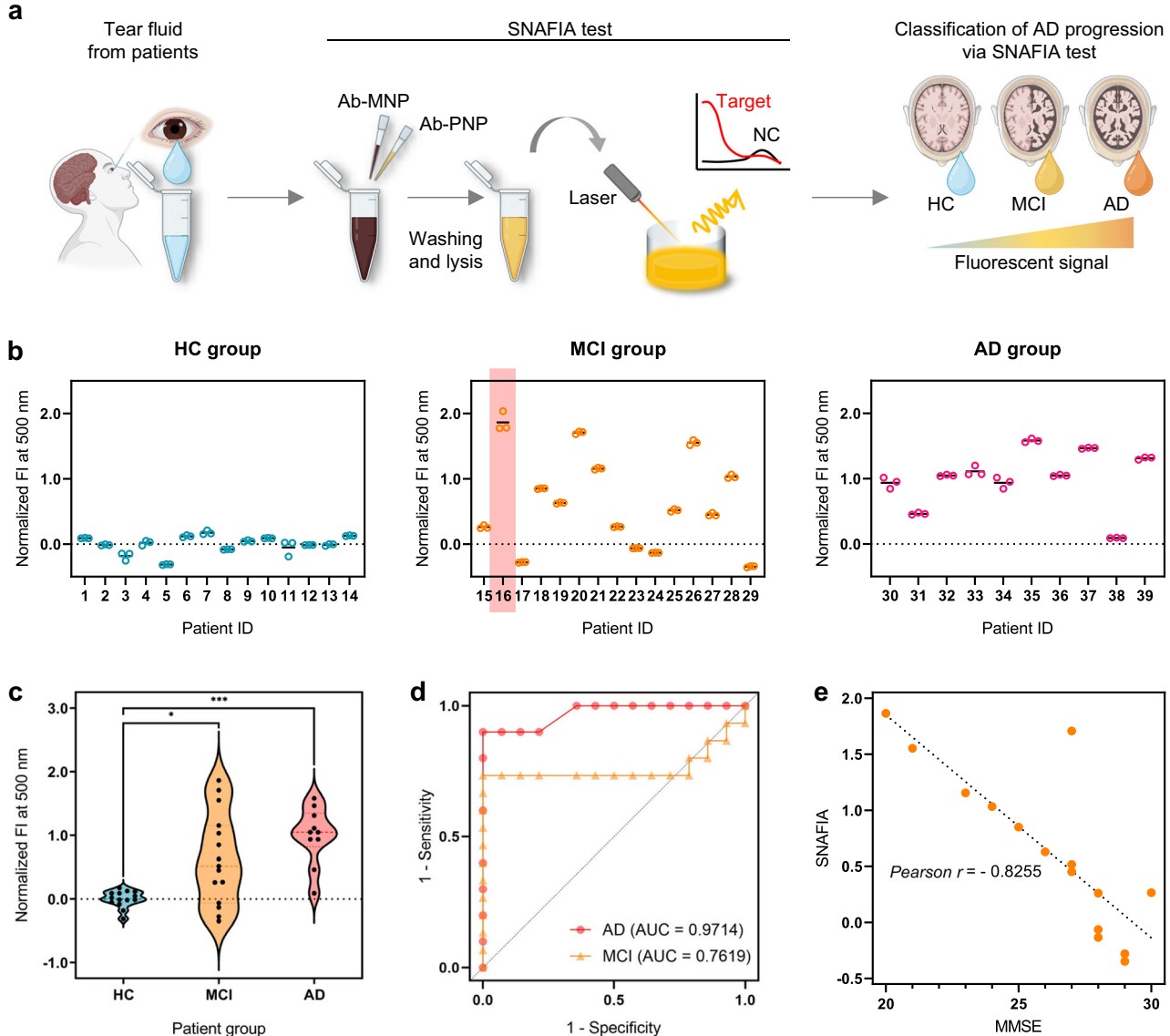

**Fig. 5 | Clinical diagnosis application of SNAFIA using tear fluids from MCI and AD patients from a verification cohort. a** Schematic of CAP1 detection by SNAFIA using tear fluid. The SNAFIA test is performed using non-invasively collected tear fluid. The intensity of the fluorescence signals obtained from the SNAFIA test determines the stage of AD progression. **b** SNAFIA test results showing the normalized FI using human tear fluid from the HC (left), MCI (middle), and AD (right) groups. The pink box labeled patient (patient ID: 16) showed disease progression from MCI to definite AD over two years. **c** Comparison of normalized FI of the CAP1 biomarker candidate in the human tear fluid of HC, MCI, and AD groups. Each signal analysis of SNAFIA was performed on HC ($n = 14$), MCI ($n = 15$), and AD ($n = 10$) individuals from the verification cohort. Statistical analysis was performed using a one-way analysis of variance (***$p = 0.0003$, *$p = 0.0251$, Kruskal-Wallis test). All measurements were performed in triplicate, and data represent mean ± s.d. **d** Receiver operating characteristic (ROC) curves of SNAFIA for MCI (yellow dots) or AD (red dots) groups compared with HC individuals. The area under the ROC curve values is shown in the graph. **e** Correlation between fluorescence signals of SNAFIA using clinical tear samples and the patient's Mini-Mental State Examination (MMSE) score. The scatter plot shows that the SNAFIA signals and MMSE scores are negatively correlated, with a Pearson correlation coefficient value of −0.8255. Statistical analysis was performed by two-tailed (***$p = 0.0002$). Schematics were created with BioRender.com.

respectively. Therefore, our nanoparticle-based immunofluorescence assay analyzed the expression of CAP1 protein in tear fluid of clinical patients with a strategy that utilized co-entrapment of target molecules using Ab-MNPs and Ab-PNPs, target-specific magnetic separation, and selective fluorescence signal amplification and showed excellent predictive results compared with traditional diagnostic methods.

AD is a neurodegenerative disease whose pathogenesis and causes are not clearly understood and may be caused by a combination of risk factors, including genetic and environmental factors. However, brain lesions are already changing in the asymptomatic stage, long before cognitive decline; therefore, it is important to discover accurate early diagnostic biomarkers and develop diagnostic technologies that can pre-diagnose patients likely to develop AD. In addition, early diagnosis of AD and identification of individual AD progression stages are expected to increase the success rate of clinical trials by pre-screening high-risk individuals, enabling personalized prevention and treatment of AD. To ensure an accurate individual diagnosis, we correlated the fluorescence intensity of SNAFIA using clinical tear samples with MMSE scores (Supplementary Fig. 19). The MMSE can describe cognitive measures based on immediate and delayed story recall of the East Boston Memory Test. In general, higher cognitive test scores on the MMSE test indicate better cognitive ability and a lower risk of AD[45]. As shown in Fig. 5e, the Pearson

correlation between SNAFIA's fluorescence intensity and MMSE score had a coefficient value (r) of −0.8255, demonstrating that SNAFIA can effectively distinguish between the prodromal and late stages of AD. Surprisingly, one of the patients diagnosed with MCI (patient ID: 16) in Fig. 5b showed disease progression to definite AD over two years, despite having a high MMSE score on initial cognitive testing. Because SNAFIA reflects an individual's current level of AD progression, our simple, reliable, and non-invasive diagnostic platform could help predict the progression to AD in patients with MCI and prevent and manage dementia through regular repeat testing with tear fluid. Taken together, SNAFIA using non-invasively collected tear fluid samples could be a feasible AD diagnostic and monitoring platform that can be repeatedly and precisely assessed for progression from MCI to AD dementia.

In conclusion, an AD diagnostic system is proposed that utilizes surface-functionalized nanomaterials (Ab-MNPs and Ab-PNPs) to induce multiple simultaneous captures, magnetic purification, and selective fluorescence signal amplification. This method enables highly sensitive and selective detection of protein biomarkers in human tear fluid. The designed SNAFIA platform exhibited a $10^7$-fold lower LOD and 10 orders of magnitude enhancement in dynamic range compared to conventional colorimetric ELISA. This greatly enhanced sensitivity was attributed to target-specific magnetic separation by Ab-MMPs in complicated samples rich in various interfering substances and amplification of the detection signal through the release of one-to-many dyes using Ab-PNPs. Indeed, the hSNAFIA test developed by replacing Ab-PNPs with fluorescently labeled antibodies showed a LOD of approximately 10 times higher than that of SNAFIA. This is due to signal bursting through selective membrane disruption with a high signal-to-noise ratio by entrapping the FRET dye molecules in the hydrophobic membrane of PNPs[46]. Using MNPs and PNPs with customized surface modification according to different target biomarkers[47,48], the SNAFIA test was able to detect AD-associated protein markers at an attomolar concentration within 1 h using a laboratory plate reader, yielding LODs of 236 and 58.6 aM for CAP1[37–39,41] and APOE[1,42–44] in liquid human biopsies, respectively. In addition to this high sensitivity, the SNAFIA assay selectively recognized CAP1 protein in human tear fluid. Finally, applying SNAFIA to 39 clinical tear samples resulted in a significant signal increase in the MCI and AD groups compared to the HC group, with an AUC value of 0.971 and 0.762, respectively. While this study serves as a proof-of-concept, it underscores the need for more extensive clinical investigations to comprehensively validate the clinical potential of this approach. If further larger-scale clinical studies confirm these findings, SNAFIA could provide a simple, fast assay with high diagnostic accuracy for a variety of protein markers using tear fluid, making it a promising tool for early diagnosis of AD.

SNAFIA is an immunodiagnostic platform that integrates a magnetic separation system and fluorescent nanoprobes, and due to the flexibility and scalability of this nanoparticle design, it has the potential to generate multiple separable optical signals targeting different biomarkers from a single sample. This multiplexing capability not only simplifies the diagnostic process but also provides test cost and time efficiencies. The cost of the SNAFIA platform depends on factors like the number of biomarkers, assay complexity, and production scale. To enhance affordability and accessibility, we aimed for a cost-effective solution. For 96 tests, MNPs and polymer nanoprobes are $130 each, while wash and lysis buffers are $5 each, totaling $270 ($2.8 per test). In comparison, the cost per test is more than three times lower than traditional immunoassays such as ELISA. With ongoing platform enhancements and mass production, we anticipate further cost reductions, making SNAFIA a practical and economical diagnostic tool for widespread clinical use. Furthermore, integration of the SNAFIA platform into intraocular lens technology could enable real-time monitoring of AD through other signal readouts, such as electrochemical signals. This extracorporeal fluid-based non-invasive measurement technique is expected to improve the issues of existing AD testing methods that are painful, invasive, and expensive, to become a system that can screen patients and high-risk groups for early diagnosis and clinical intervention of AD development.

## Methods

### Ethical Statement
This study was approved by the Institutional Ethical Review Boards of Yonsei University College of Medicine (Seoul, South Korea; IRB No. 3-2018-0156). Informed consent was obtained from all subjects or their authorized representatives.

### Materials
The following chemicals were purchased from Sigma-Aldrich (Saint Louis, MO, USA): Ammonium bicarbonate, iron trichloride, sodium citrate, sodium acetate, ethylene glycol, diethylene glycol, tetraethyl orthosilicate, 1-ethyl-3-(3-(dimethylamino)-propyl) carbodiimide, N-hydroxysulfosuccinimide, anti-goat IgG (whole molecule)-gold antibody produced in rabbit, methoxy polyethylene glycol (mPEG) with a molecular weight of 2 kDa, D, L-lactide (3,6-dimethyl-1,4-diox-ane-2,5-dione), tin (II)-ethyl hexanoate, anhydrous chloroform, and Triton X-100. C18 Harvard macro spin column was obtained from Harvard Apparatus (Holliston, MA, USA). Silane PEG acid with a molecular weight of 2 kDa was obtained from Nanocs (NY, USA). Tandem mass tag (TMT) isobaric mass tagging reagent, C18 LC column, C18 spin column, Q Exactive orbitrap hybrid mass spectrometer, 3,3'-dioctadecyloxacarbocyanine perchlorate, 1,1'-dioctadecyl-3,3,3',3'-tetramethylindocarbocyanine perchlorate, BupH™ MES buffered saline, BupH™ Phosphate buffered saline, and Pierce™ BCA protein assay kit were purchased from Thermo Fisher Scientific (Waltham, MA, USA). Poly(D, L-lactide)-PEG-methyl ether (PDLLA-mPEG) with a molecular weight of 7 kDa/2 kDa of PDLLA/PEG and Poly(D, L-lactide)-block-poly(ethylene glycol)-carboxylic acid (PDLLA-PEG-COOH) with a molecular weight of 7 kDa/2 kDa of PDLLA/PEG average molecular weight of 2 kDa of PEG were obtained from Nanosoft Polymers (Winston-Salem, NC, USA). A cellulose ester (CE) membrane (Spectra/Por® Biotech CE Tubing, MWCO: 10 kDa, 300 kDa) was purchased from Spectrum Labs (Los Angeles, CA, USA). Apolipoprotein E (APOE) protein (cat. no. ENZ-PRT263-0500) was purchased from Enzo. A human APOE antibody pair (cat. no. ab244096) and a human APOE SimpleStep ELISA kit (product no. ab233623) were purchased from Abcam (Cambridge, UK). The antibody pair antibodies were rabbit monoclonal and primary antibodies. Adenylate cyclase-associated protein 1 (CAP1) protein (cat. no. LS-G74326), CAP1 antibodies (cat. no. LS-C381814 and cat. no. LS-C411294), and a human CAP1 ELISA kit (product no. LS-F22718) were purchased from LSBio. The CAP1 antibodies were rabbit polyclonal and primary antibodies. ABflo® 488-conjugated Goat Anti-Rabbit IgG (cat. no. AS053) was a secondary antibody and was purchased from ABclonal. All other chemicals and reagents were of analytical grade and used without further purification unless otherwise stated. 400-mesh copper grids were purchased from Ted Pella, Inc. (Redding, CA, USA).

### Human tear sampling
A total of 60 participants were prospectively enrolled and conducted at the Memory Disorder Clinic of Gangnam Severance Hospital (Seoul, Korea) from July 2018 to June 2019 with approval from the institutional review board (IRB no. 3-2018-0156). Participants underwent neuropsychological testing and brain imaging, including ¹⁸F-florbetaben PET scans[49]. AD and MCI were diagnosed by a neurologist (H.C) based on the clinical diagnoses and amyloid-β positivity[50,51]. The study was designed to incorporate two distinct human cohorts: a discovery cohort comprised of 7 HC, 7 MCI, and 7 AD participants; and a verification cohort consisting of 14 HC, 15 MCI, and 10 AD participants.

Applying our proteomics analysis to the power analysis data[52,53], the minimum sample size required for a power analysis would be 15 for MCI and 10 for AD when compared to HC. Detailed demographics are summarized in Supplementary Tables 1 and 10. Patients with dry eye disease were excluded, as it can influence the protein composition of tears. Those with any history of ocular abnormalities in either eye, including ocular surgery, injury, infection, allergy, inflammation (such as uveitis), glaucoma, or retinal diseases were also excluded, as these conditions can introduce variations in tear composition unrelated to AD or MCI. Patients were not included using any topical eyedrops other than artificial tears and contact lens wearers. Additionally, individuals with systemic diseases like autoimmune disorders, diabetes, and vascular diseases, which can have systemic effects were not part of our study. To perform global proteome profiling for the discovery phase and SNAFIA assay for the verification phase, tear fluid was collected from participants using a bonded $2.0 \times 10$ mm polyester fiber rod (TRANSORB® WICKS, FILTRONA, VA, USA)[54]. The polyester wick was applied to the tear meniscus at the lower eyelid margin. It was then placed in a 1.5 mL Eppendorf (EP) tube and absorbed tear fluid was recovered from the fiber by batch centrifugation for each subject at $2200 \times g$ for seven minutes using an EP centrifuge (Westbury, NY). The retrieved tear proteins were stored at −70 °C until a mass spectrophotometric assay was performed.

## Proteomics analysis process for human tear biomarker discovery

As shown in Fig. 2a, proteins (100 µg) of the pooled tear fluids from each group were placed in duplicate in two tubes and digested into peptides by in-solution digestion. In detail, 8 M urea in 100 mM ammonium bicarbonate was mixed with each fluid sample at a 1:1 ratio, and incubated for 30 min at room temperature (RT). Samples were reduced with 10 mM dithiothreitol at RT for 30 min, followed by 30 mM iodoacetamide alkylation in the dark for another 45 min. Trypsin was added to the samples at a volume ratio of 1:50 and incubated at 37 °C overnight. The activated trypsin reaction was quenched with 0.8% trifluoroacetate, and peptides were desalted with a C18 Harvard macro spin column. The resultant peptides were dried in a Speed-Vac concentrator and stored at −80 °C.

Each peptide sample was labeled with TMT isobaric mass tagging reagents according to the manufacturer's instructions. The chemically tagged samples were pooled into one tube, and peptide separation was performed using high pH reverse-phase liquid chromatography fractionation using the Q Exactive Orbitrap hybrid mass spectrometer coupled with the nanoAQUITY UPLC system (Fig. 2a).

Full mass spectrometry data were acquired at a resolution of 70,000 at m/z 200 in a scan range of 400−2000 Th with an automated gain control target value of $1.0 \times 10^6$ and a maximum ion injection of 120 ms. DEPs as biomarker candidates in MCI and AD compared with those in HC were defined as proteins with a fold change of greater than two and a p-value less than 0.01. A gene ontology search was performed to explore the biological processes related to the tear DEPs associated with MCI and AD (p-value < 0.01). Protein–protein interactome information was collected from the STRING public database (version 11.5) to build a network showing enriched processes. A network model was built from tear DEPs and interactome data using Cytoscape software (version 3.7.2)[55].

## Quantitative global proteome profiling and data processing for human tear biomarker discovery

The obtained MS/MS spectra were analyzed using the Proteome Discoverer software version 2.1 (Thermo Scientific), searching against the UniProt human database with the SEQUEST HT® search engine. The static modifications included carbamidomethylation (C) and TMT six-plex (N-terminal, lysine (K)). In contrast, dynamic modifications encompassed methionine oxidation. Using the Percolator algorithm, the resulting peptide hits were filtered to ensure a maximum false discovery rate (FDR) of 1%.

For quantification, the TMT 6-plex method was explored within the Proteome Discoverer software, calculating the reporter ratios with a mass tolerance of ±10 ppm. Subsequently, the protein intensities were normalized by total sum scaling to ensure comparability. The ratio of mean values was calculated of normalized protein intensities for each protein and a p-value was determined for each protein using a t-test. DEPs were defined in MCI and AD compared with HC based on a fold change greater than 1.5 and less than 0.67 and p-values less than 0.01 (Supplementary Table 2).

## Synthesis of magnetic nanoparticles (MNPs) and antibody-immobilized MNPs (Ab-MNPs)

For the synthesis of iron oxide nanoparticles, iron trichloride (0.35 g), sodium citrate monobasic (0.15 g), and sodium acetate (1.20 g) were dissolved in 20 mL of a cosolvent (diethylene glycol: ethylene glycol = 3:1% (v/v)), stirred with a magnetic bar for 30 min, and sealed in a Teflon-lined stainless-steel autoclave. After heating at 220 °C for 12 h and cooling to 25 °C, the resulting MNPs were washed five times with ethanol and deionized water (DW), respectively. The MNPs with a silica shell (SiMNPs) were synthesized using the modified Stöber method. 10 mL of MNP solution (9 mg/mL) was diluted with 40 mL of ethyl alcohol and 1.12 mL of ammonia solution. Next, 0.5 mL of tetraethyl orthosilicate (TEOS) was added to the reaction mixture and allowed to react under mechanical stirring with a glass impeller for 12 h at 25 °C. During the reaction, silica shells formed on the MNPs' surface via hydrolysis and condensation of TEOS. Finally, the synthesized SiMNPs were magnetically collected and washed thrice with ethanol and DW.

For conjugation with antibody, the surface of SiMNPs (10 mg) was functionalized with carboxylic acid groups by incubating with 500 µL of silane-polyethylene glycol-carboxylic acid solution (2 kDa, 50 mg/mL) for 6 h under vortexing vigorously at 25 °C. After incubation, the carboxylated MNPs (CMNPs) were rinsed three times with DW. Finally, antibody-conjugated MNPs (Ab-MNPs) were obtained by incubating 3 mg of the CMNPs with 50 mM of 1-ethyl-3-(3-(dimethylamino)-propyl) carbodiimide (EDC) and N-hydroxysulfosuccinimide (sulfo-NHS) in 3 mL of MES buffer (0.1 M MES, 0.15 M NaCl, pH 4.5) using the carbodiimide method. After activating for 15 min at 25 °C, the CMNPs were separated using a magnetic separator (Invitrogen) and washed with 1× PBS buffer (pH 7.4) three times. Finally, the CMNPs were dispersed in 3 mL of 1× PBS buffer (pH 7.4) and were covalently coupled with 50 µL of a capture antibody of human APOE antibody pair (cat. no. ab244096; Abcam) and CAP1 antibody (cat. no. LS-C411294; LSBio) for 2 h at 25 °C, respectively. The antibody solutions were prepared at 1 mg/mL concentration in 1× PBS buffer (pH 7.4). The unbound antibody was washed off by magnetic separation, and the resulting Ab-MNPs were dispersed in 3 mL of 1× PBS buffer (pH 7.4).

## Synthesis of methoxy polyethylene glycol-*block*-poly lactic acid (mPEG-*b*-PLA) copolymers

Methoxy polyethylene glycol-block-polylactic acid (mPEG-*b*-PLA) copolymers were synthesized by ring-opening polymerization of D, L-lactide monomers with hydroxyl-terminated mPEG (2 kDa) as an initiator and tin (II)-ethyl hexanoate (Sn(Oct)$_2$) as a catalyst[20,25]. One gram of mPEG was introduced into a three-neck flask along with anhydrous toluene. Four grams of D, L-lactide and 10 µL of Sn(Oct)$_2$ (0.05% (w/w) solution in toluene) were then injected into the flask, and the reaction mixture was heated with reflux at 120 °C overnight under a nitrogen atmosphere. After the reaction was completed, the product solution was removed by rotary evaporation and precipitated in excess cold diethyl ether to produce mPEG-*b*-PLA. The copolymer products were filtered using a Buchner funnel and vacuum-dried at 18 °C for a day. The molecular weight and PLA content of the synthesized mPEG-*b*-PLA were determined from proton nuclear magnetic resonance

(¹H NMR) spectra with chemical shift values of 3.65 ppm (–CH₂– of PLA) and 3.38 ppm (–OCH₃– of mPEG) using a 400 MHz ¹H NMR spectrometer (Bruker, Bremen, Germany) with deuterated chloroform as solvent and TopSpin software (version 3.5). Their chemical structures were analyzed using Fourier-transform infrared spectroscopy (Excalibur Series, Varian Inc., Palo Alto, CA, USA) and Spectrum software (version 10.3.6) with the presence of a characteristic peak at 1750 cm⁻¹ representing carbonyl stretching (C = O) of the ester in mPEG-*b*-PLA copolymers.

## Preparation of antibody-immobilized polymeric nanoprobes (Ab-PNPs)

PNPs were prepared using a thin film hydration and tip sonication method. mPEG-*b*-PLA copolymer (7.5 mg) and COOH-PEG-*b*-PLA (2.5 mg) were dissolved in 1 mL of chloroform to prepare 10 mg/mL of polymer solution and mixed with 20 μL of 3,3'-dioctadecylox-acarbocyanine perchlorate (DiO) and 1,1'-dioctadecyl-3,3,3',3'-tetra-methylindocarbocyanine perchlorate (DiI) solutions (1 mg/mL in chloroform). The chloroform was eliminated by operating a vacuum rotary evaporator for 30 min to form thin films on the bottom of round-bottom flasks. The thin films were then entirely hydrated with 5 mL of DW by incubating at 60 °C for 6 h and magnetic stirring at 60 °C overnight. The dye-containing PNP dispersion was sonicated using a tip sonicator (VCX-750 Vibra Cell Processor, Sonics & Materials, Inc. Newtown, CT, USA) for 10 min to make the PNPs homogeneous and monodispersive. The prepared PNP solution was then placed into cellulose ester dialysis membrane tubing (molecular weight cutoff 10 kDa) and dialyzed against DW under mild stirring for 48 h to remove residual dyes. Antibody-conjugated PNPs (Ab-PNPs) were obtained by incubating 1 mL of PNPs solution with 50 mM of EDC and sulfo-NHS in 3 mL of MES buffer (0.1 M MES, 0.15 M NaCl, pH 4.5) using the carbo-diimide method. After activating for 15 min at 25 °C, the activated PNPs were covalently coupled with 50 μL of a detector antibody of human APOE antibody pair (cat. no. ab244096; Abcam) and CAP1 antibody (cat. no. LS-C381814; LSBio) for 2 h at 25 °C, respectively. The antibody solutions were prepared at 1 mg/mL concentration in 1× PBS buffer (pH 7.4). The unbound antibodies were purified by cellulose ester dialysis membrane tubing (molecular weight cutoff 300 kDa) and dialyzed against DW under mild stirring for 48 h.

## Characterization of Ab-MNPs and Ab-PNPs

High-resolution TEM images for investigating the morphology of nanoparticles were obtained using a JEOL JEM-F200 (JEOL, Tokyo, Japan) operating at an acceleration voltage of 200 kV, equipped with an energy-dispersive X-ray spectroscopy (EDS) detector and analyzed a DigitalMicrograph software (version 3.0) and Aztec software (version 6.0). The TEM samples were prepared by placing 20 μL of nanoparticle solutions on the 400-mesh copper grids (Ted Pella, USA). Before TEM imaging, the Ab-PNP samples were negatively stained using a 3% (w/v) phosphotungstic acid solution (pH 6.8) and dried overnight. The magnetic properties of MNPs and Ab-MNPs were examined using a vibrating sample magnetometer (VSM; MODEL-7407, Lake Shore, OH, USA) at 25 °C, and the curves were obtained using Lake Shore IDEAS VSM software (version 3.6). Powder X-ray diffraction (XRD) patterns were recorded using a Rigaku Ultima III X-ray diffractometer (Rigaku, MD, USA) and processed using JADE software (version 5.0). The VSM and XRD samples were prepared by lyophilizing the MNPs and Ab-MNPs solutions into powder. Dynamic light scattering (DLS) and Zeta potential were obtained using an ELSZ-2000ZS (Otsuka Electronics, Osaka, Japan) at 25 °C with data obtained in triplicate measurements for 0.1 mg/mL of sample solutions. All fluorescence and absorbance were obtained using the SpectraMax® i3x multimode microplate reader (Molecular Devices, CA, USA) at 25 °C and analyzed using SoftMax Pro software (version 7.0.3). To maximize the FRET effect, the fluorescent signal in the supernatant is quantitatively analyzed by

recording the emission fluorescence spectra between 500 and 650 nm upon excitation at 475 nm using the SpectraMax i3x multimode microplate reader at 20 °C. The FRET ratio and FRET change efficiency were calculated using the fluorescence intensity of DiO ($F_{DiO}$) and DiI ($F_{DiI}$), as depicted in Eq. 1 and Eq. 2, respectively.

$$FRET\ ratio = F_{DiI}/(F_{DiO} + F_{DiI}) \tag{1}$$

$$FRET\ change\ efficiency\ (\%) = 1 - \{F_{DiI}/(F_{DiO} + F_{DiI})\} \times 100 \tag{2}$$

## SNAFIA test for the detection of target biomarkers

Ab-MNPs solution (100 μL) and Ab-PNPs solution (100 μL) were mixed with 100 μL of the sample solution in a 1.5 mL EP tube. For the SNAFIA test using clinical samples from the verification cohort, 8 μL of the human tear sample was dispersed in 92 μL of artificial tear fluid and prepared in advance. The mixture was incubated at 37 °C for 30 min under 7 × g vortexing using a thermo-shaker (MS-100 Thermo Shaker, TAESHIN BioScience, Gyeonggi-do, Korea) to form a sandwich immunocomplex. The Ab-MNPs and Ab-PNPs bound to the target were separated by a magnetic separator and washed three times with 1× PBS buffer (pH 7.4). Next, 100 μL of 1% (w/w) TX-100 solution was treated and incubated at 37 °C for 15 min under 7 ×g vortexing using a thermo-shaker. Finally, the immunocomplexes were magnetically separated from the mixture, and the supernatants containing DiO and DiI dyes released from the Ab-PNPs were transferred to a 96-well amber microplate. To assess the concentration of target proteins, the fluorescent signal in the supernatant is quantitatively analyzed by recording the emission fluorescence spectra between 500 and 650 nm upon excitation at 475 nm using the SpectraMax i3x multimode microplate reader at 20 °C. The fluorescence intensity was normalized to the maximum intensity of the solution at 500 nm and used as a signal readout. The data were fitted to a four-parameter logistic curve, and the LOD was calculated using the three-sigma (3σ) method. LOD was obtained using the standard deviation of the blank (σ), the slope (S) of the linear regression, and the mean value of the blank ($\mu_B$), as shown in Eq. 3, and the result was substituted into the linear regression.

$$LOD = 3\sigma + \mu_B \tag{3}$$

## ELISA test

Standards of target proteins, including APOE (cat. no. ENZ-PRT263-0500; Enzo) and CAP1 (cat. no. LS-G74326; LSBio), were prepared at a series of concentrations in 1× PBS buffer (pH 7.4). Enzyme-linked immunosorbent assay (ELISA) tests for the detection of target APOE and CAP1 proteins were conducted according to the manufacturer's instructions using a human APOE ELISA kit (product no. ab233623; Abcam) and a human CAP1 ELISA kit (product no. LS-F22718; LSBio), respectively. Both kits were pre-coated with capture antibodies, and all reagents were prepared according to the instructions. In the APOE ELISA kit, 50 μL of standard or sample was added to each well of the microplate, then 50 μL of antibody cocktail was added to the wells and incubated at 25 °C for 1 h. Each well was washed three times with 250 μL of 1× wash buffer, followed by adding 100 μL of 3,3',5,5'-tetra-methylbenzidine (TMB) development solution and incubated for 10 min. The color produced by the TMB substrate was initially blue and then turned yellow upon the addition of the stop solution. The colored product was detected using a SpectraMax i3x multimode microplate reader at an absorbance of 450 nm. Similarly, for the CAP1 ELISA kit, 100 μL of standard or sample was added to each well, followed by adding 100 μL of 1× biotinylated detection antibody and incubating at 37 °C for 1 h. Each well was washed five times with 250 μL of 1× wash

buffer, followed by adding 90 μL of TMB substrate solution and incubating at 37 °C for 15 min. After adding 50 μL of stop solution, the absorbance was immediately measured at 450 nm.

## Half-SNAFIA (hSNAFIA) test

For a sensitivity comparison with SNAFIA, the formed immunocomplex between Ab-MNPs and the target analyte was labeled with secondary antibodies instead of Ab-PNPs. Immunocomplex formation composed of Ab-MNPs, target proteins, and primary capture antibodies was carried out the same as described for the SNAIFA test. The immunocomplexes were separated with a magnetic separator and washed three times with 1× PBS buffer (pH 7.4). Next, 100 μL of secondary antibody solution conjugated with AlexaFluor®488 (cat. no. AS053; ABclonal) and 50-fold diluted in 1× PBS buffer (pH 7.4) was added and incubated at 37 °C. After 30 min, the mixture was washed three times with a magnetic separator, followed by treatment with 100 μL of 1% (w/v) TX-100 solution at 37 °C for 15 min. The resulting supernatants were transferred to a 96-well microplate, and the fluorescence intensity (excitation 475 nm, emission 520 nm) was measured using a microplate reader.

## Statistics and reproducibility

All graphs in this study were drawn using GraphPad Prism 9 software (version 9.0.0). Diagnostic performance (e.g., ROC curve, LOD) and statistical analysis (Pearson's coefficient, $p$-values) of SNAFIA were performed using SPSS 23 (version 17.0.2) and GraphPad Prism 9 software. Image J (version 23) was used for image-based nanoparticle size measurement. Results are presented as mean ± S.D. The t-tests with a 95% confidence interval were used to determine the significance of the data between the two groups. One-way analysis of variance was conducted to determine the significance of data with more than two groups and was followed by Dunn's test, Brown-Forsythe and Welch, Mann-Whitney test, and Brown-Forsythe and Barlett's tests. No statistical method was used to predetermine the sample size. Throughout the study, the sample size was determined based on our preliminary studies and on the criteria in the field. At least three biological samples were included for one experiment and one to three independent experiments were performed to ensure sufficient reproducibility of the results. Biological replicates (N) and the numbers of the independent experiments are indicated in each figure legend.

## Reporting summary

Further information on research design is available in the Nature Portfolio Reporting Summary linked to this article.

# Data availability

All quantified proteins are provided in Supplementary Data 1. The mass spectrometry proteomics data have been deposited to the ProteomeXchange Consortium via the PRIDE repository with the dataset identifier (PXD042142). The main data supporting the findings of this study are available in the paper and Supplementary Information. All raw and analyzed datasets generated during this study are provided as a Source Data file. Source data are provided with this paper.

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

## Acknowledgements

S.H. acknowledges support from the National Research Foundation of Korea(NRF) grant funded by the Korea government(MSIT) (No. RS-2023-00236421) and by a grant of the Korea Health Technology R&D Project through the Korea Health Industry Development Institute (KHIDI), funded by the Ministry of Health & Welfare, Republic of Korea (No. HI18C1159000020). Y.W.Ji. acknowledges support from the Basic Science Research Program through the National Research Foundation of Korea (NRF-2021R1A2C4001596) and the faculty research grant of Yonsei University College of Medicine (No. 6-2021-0117). E. K. acknowledges support from the National Research Foundation of Korea (NRF) grant funded by the Korea government (MSIT) (No. 2022R1C1C1005390).

## Author contributions

H.Cho., Y.W.Ji., and S.H. conceived the project. S.L., E.K., C.-E.M., C.P., J.-W.L., M.B., M.-K.S. and J.K. designed the experiments. S.L. and M.-K.S. synthesized MNPs and performed the experiments. S.L. and C.P. synthesized the polymer and PNPs. M.B. and H.C. prepared and analyzed clinical samples. C.-E.M. and Y.W.Ji. conducted a proteomics analysis using tear fluid. S.L. designed and performed the SNAFIA test. All authors analyzed the results. S.L. drafted the manuscript. E.K. and Y.W.Ji. edited the draft. All authors reviewed and commented on the paper.

## Competing interests

The authors declare no competing interests.
