## [Peer Review File · Nature Communications]

REVIEWER COMMENTS

Reviewer #1 (Remarks to the Author):

The authors present a classical workflow for biomarker discovery and the establishment of SNAFIA, a highly sensitive test for the examination of very low concentrations of proteins in biological fluids.

The data shown in the manuscript are of high interest, however the balance between the presentation of proteomics experiment, which is the leading trait of the story and the meticulous presentation of how the SNAFIA test was generated is shifted toward the later.

Based on the title, abstract and introduction I would wait for the presentation of how the selected proteins can discriminate between patients with AD and controls. Instead, a very detailed, too technical presentation of the system used for SNAFIA is shown. My feeling is that regarding the presentation of the data, the decision to be a technical note or a biomarker discovery story was not made.

In the current form it is not clear why APOE and CAP1 proteins were selected and how do they come to the story. My recommendation is to restructure the results and discussion part: start with the presentation of the proteomics experiments, its readouts and the justification of the necessity of the verification of APOE and CAP1. After this I would move to the verification part and I would highlight the biological (and analytical) power of the SNAFIA test and show much less details regarding the generation of the nanoparticles used for the test. All the details regarding the particle generation and testing (current Figure 1 and 2) I would move to the supplementary materials. According to my opinion in the current form, they do not fit into the main text. It seems to me as two manuscripts: i) a technical description of SNAFIA generation (Fig. 1, 2, Suppl. Figures 1-9) and ii) identification of potential biomarkers for AD (Fig. 3-5, Suppl. Figures 10-17) would be mixed.

Shortcomings:

the title and the abstract suggest a biomarker discovery story, but the reader has to cope first with a very detailed, technical presentation of the nanoparticle generation and testing. The too many details in the main text divert the attention from the message of the manuscript. The detailed presentation of SNAFIA generation is important but I would suggest either presenting in a different article or in the supplementary part.

it is not listed how many samples were used for tear pool generation

please include how the tear proteins were retrieved from the thread

the number of samples (39) included into the analysis is low and for verification this is too low number

the biomarker discovery and the verification of selected potential markers was done on the same samples decreasing the power of the results

it is not clear why APOE was selected for study

very few references are used in the discussion part

Points to be addressed

According to the current knowledge not APOE itself, but APOE e4 allele is a risk factor for AD. Please correct this in the text.

Please give information on which APOE is recognized by the antibody. APOE4 or all APOEs?

Sentence in line 299 should be modified. There is no presented evidence that "SNAFIA has the ability to directly detect AD biomarkers as low as 0.06 fM". The data refer only to APOE.

Please show results for APOE spiked in tears. What is the dilution factor for tears?

Please show results for the concentration measurement of APOE in tear and serum of controls and/or patients. I think it is important to demonstrate that the test performs well with real biological fluids.

I think the authors do not have enough evidence to make the following statement: "This demonstrates that the SNAFIA platform has excellent selectivity, even in complex

samples with non-specific proteins.” Having combined two proteins is very far from the complex biological samples. Please modify the statement accordingly.

It is not clear why APOE was tested in serum and PBS and CAP1 in artificial tear? Please perform experiments using both SNAFIA for APOE and CAP1 for PBS, serum, artificial tear and pooled tear from patients. Please show the results of spiked proteins into these matrixes and the amount quantified in the serum and tear.

Based on which data the authors define progressively up and gradually downregulated proteins?

Please also comment in the text the components of chemical barrier having role in defense as shown on figure 4 b. There are available information on the changes of the components of the chemical barrier and the network of antimicrobial and immunomodulatory peptides in tear and in AD in the literature. I think it is important to highlight besides the other functions these functions as well.

Why results with APOE are shown if only CAP1 SNAFIA was used later? It is not clear why APOE was included if CAP1 is the potential biomarker.

According to the presented data CAP1 does not fulfill all the requirements for a biomarker, please refer to it in the text as potential biomarker instead of biomarker. Only after validation on large patient number (thousands of patients) we can call a protein biomarker. The use of “biomarker” term in the text should be revisited and used accordingly.

In the sentence “SNAFIA test was able to accurately identify the MCI and AD patient groups in the HC participants” the preposition in is inappropriate. Please change accordingly.

Please change the scales to log scale on Suppl. Fig. 11. and 15

Please clarify what is the difference between the patients shown in Suppl. Table 5 and 9. In the methods where the tear collection is described Suppl. Table 5 is mentioned, while in the text Suppl. Table 9.

Please give a detailed description on how the proteomics analysis was carried out. A detailed description of the sample preparation and analysis: digestion, more details on TMT labeling, sample cleanup,

chromatographic separation, fractionation, mass spec conditions (settings), type of data acquisition method applied is required.

Please describe which software were used and how the mass spectrometry data were processed. Please indicate the database version, applied settings and the FDR correction of the data.

Please show a list of quantified proteins in Supplementary table.

Please indicate the statistical analysis used for data evaluation and show the list of DE proteins in a supplementary table and volcano plot in the main text.

Please indicate why CAP1 protein was selected for further analyses. This had the highest change compared to controls or based on some other features?

Please upload the proteomics data to public repository, such as PRIDE or ProteomeXchange (<https://www.proteomexchange.org/>) and indicate the accession number in the abstract and in the main text.

Please give details on the network generation. What stringency, value was applied, which String db version was used and please add reference for String DB.

Please comment on the multiplexing possibility and price of SNAFIA.

Please comment based on literature data the CAP1 protein. Was it discovered as a potential biomarker in other diseases, is it normally present in tears and other body fluids, etc. ?

Reviewer #2 (Remarks to the Author):

The fluorogenic immunoassay designed by the authors and the exhaustive controls carried out, do this MS interesting by itself. They demonstrate that is a useful technique to detect proteins in tears. Taking into account its sensibility, this technique can be useful for early diagnosis and monitoring of

Alzheimer's disease from tear fluid or any other ophthalmic problem. However, to affirm that CAP-1 detected by this technique can be used as a biomarker for AD are necessary more additional data.

Thus,

1. The mass spectrometry proteomics data have to be deposited in a public repository.
2. Information with respect to the time to obtain the tear samples, tear volume and more important the presence of ophthalmic problems such as ophthalmic allergy, ophthalmic infection, ocular hypertension, Dry eye, and/or cataracts have to be published. AD and MCI used to have these problems.
3. ApoE4 genotype for human samples should be given
4. To ensure the quality of the samples, the presence of well-established tear markers such as lysozyme, lipocalin 1, serotransferrin or retinal dehydrogenase 1 should be published, This is important because slight epithelial damage caused by sampling is always taking place
5. The authors talk about "the critical role of CAP1 pathogenesis AD", but only a reference is given and related to its presence in tear samples.
6. It is confusing to test the technique against ApoE (authors present many experimental controls) or AchEase and not to test them in human samples. If there are no changes, that is informative. With respect to ApoE could be interesting to test if, taking into account the genotype, a decrease or increase in ApoE is observed.
7. A table with all the proteins analyzed should be shown, at least in a supplementary table and with the name of them.
8. Fig. 5B: Data should be corrected by protein content or at least by volume of tear. I understand that this correction has not been performed.
9. Authors represent CAP-1 levels vs MMSE. Taking into account that they have also performed brain imaging, including 18F- florbetaben positron emission tomography scans could be interesting to do the same representations

Minor:

1. Fig4B, explain what grey circles means
2. Could be expressed data Fig 5B in absolute terms? You can do a calibration curve with CAP-1 and then express the concentration of CAP1 in your sample.
- 3- Fig 5b (MCI group): Pink box is explained in line 480, but should be explained in the figure legend).

Reviewer #4 (Remarks to the Author):

The authors report a new FRET sensing platform for the rapid detection of AD biomarkers in tears. It is based on magnetic nanoparticles functionalized with the detection antibodies. The article starts from the synthesis and characterization of the NPs, continues with the sensing principles and finishes with the tests performed on real samples.

The sensing principle and the functionalization procedure are not novel. But the biological target is original and of utmost importance for the early diagnosis of AD. The analytical performances are remarkable. The diagnosis could be based on the identification of proteins in tears with a very good sensitivity. How does it compare with digital ELISA such as the Simoa platform? The obtained results are convincing; the manuscript is clear and well-written. Even if the sensing strategy is not completely novel, considering the quality of the bioanalytical results, the importance and the potential medical and societal impact of the reported immunoassays, I support the publication of this manuscript in Nature Communications.

Minor point:

-Fig. 2e : it should be « Immunolabeled... » instead of « Immunolabed... »

Response to Reviewers' Comments

We sincerely appreciate the reviewers for their time and their constructive criticism of our manuscript. It has undoubtedly improved our manuscript as a result of their contributions. Below we provided our responses to the points raised. The point-by-point responses are in blue, and changes to the manuscript are highlighted in red.

Reviewer #1:

The authors present a classical workflow for biomarker discovery and the establishment of SNAFIA, a highly sensitive test for the examination of very low concentrations of proteins in biological fluids.

The data shown in the manuscript are of high interest, however the balance between the presentation of proteomics experiment, which is the leading trait of the story and the meticulous presentation of how the SNAFIA test was generated is shifted toward the later.

Based on the title, abstract and introduction I would wait for the presentation of how the selected proteins can discriminate between patients with AD and controls. Instead, a very detailed, too technical presentation of the system used for SNAFIA is shown. My feeling is that regarding the presentation of the data, the decision to be a technical note or a biomarker discovery story was not made.

In the current form it is not clear why APOE and CAP1 proteins were selected and how do they come to the story. My recommendation is to restructure the results and discussion part: start with the presentation of the proteomics experiments, its readouts and the justification of the necessity of the verification of APOE and CAP1. After this I would move to the verification part and I would highlight the biological (and analytical) power of the SNAFIA test and show much less details regarding the generation of the nanoparticles used for the test. All the details regarding the particle generation and testing (current Figure 1 and 2) I would move to the supplementary materials. According to my opinion in the current form, they do not fit into the main text. It seems to me as two manuscripts: i) a technical description of SNAFIA generation (Fig. 1, 2, Suppl. Figures 1-9) and ii) identification of potential biomarkers for AD (Fig. 3-5, Suppl. Figures 10-17) would be mixed.

We appreciate the reviewer for their time and effort in providing detailed and constructive feedback on the manuscript. The reviewer raises important concerns that emphasize the need for a balanced presentation of our study. The primary objective of our study is to introduce the development of a diagnostic tool that can rapidly and sensitively detect potential protein biomarkers, thereby facilitating early diagnosis and ongoing monitoring of Alzheimer's disease (AD). As pointed out by the reviewer, we agree that the original manuscript was dominated by a detailed and technical description of the SNAFIA technology rather than a discussion of biomarker discovery and selection through proteomic analysis. Therefore, we recognized the need to address the results and discussion of

potential biomarker discovery and selection and reorganized the *Results and Discussion* section of the manuscript based on the reviewer's comments.

In the revised manuscript, we first describe the collection of patient tear fluid samples for the discovery and validation cohort studies. Next, we present and analyze the results of the proteomics experiments performed in the discovery cohort, which led to the selection of potential biomarkers. To conduct the validation cohort, we describe the design and brief technical characteristics of the SNAFIA test platform for the detection of the chosen biomarkers. Finally, we added results and discussions of the analytical performance of SNAFIA and its diagnostic capability on clinical samples.

We believe that the revised manuscript now clearly presents the main purpose and results of this study, and we hope that the responses detailed below address the reviewer's points and satisfy the concerns raised.

Shortcomings:

the title and the abstract suggest a biomarker discovery story, but the reader has to cope first with a very detailed, technical presentation of the nanoparticle generation and testing. The too many details in the main text divert the attention from the message of the manuscript. The detailed presentation of SNAFIA generation is important but I would suggest either presenting in a different article or in the supplementary part.

We thank the reviewer for the insightful comments. Based on the reviewer's feedback, we have relocated the comprehensive and technical description of nanoparticle testing to the *Supplementary Information*. We hope that the revised manuscript now clearly communicates the purpose of our study.

Q1. it is not listed how many samples were used for tear pool generation

We thank the reviewer for this point. In this study, we utilized two distinct cohorts of human tear samples for our proteomics experiments: **a discovery cohort and a verification cohort**. All subjects received ¹⁸F-florbetaben positron emission tomography (PET) imaging for amyloid-β measurement. Specifically, **the discovery cohort** comprised 21 samples, including tear samples from 7 healthy controls (HC), 7 patients with mild cognitive impairment (MCI), and 7 patients with Alzheimer's disease (AD). The **verification cohort** included tear samples from 14 HC, 15 MCI, and 10 AD patients for **a total of 39 samples**.

Thus, the study provided a robust dataset for analysis with a total of 60 tear samples. We believe this sample size is appropriate for the scope and objectives of our study and clarified this information in the revised manuscript to avoid any ambiguity.

Revised Fig. 1:

Fig. 1 Workflow of a discovery and verification cohort study for early diagnosis and monitoring of Alzheimer's disease. Sixty human tear fluid samples were utilized in two cohorts: a discovery cohort and a verification cohort. The discovery cohort was comprised of tear samples from 7 healthy controls (HC), 7 patients with mild cognitive impairment (MCI), and 7 patients with Alzheimer's disease (AD) for a total of 21 tear samples. Proteomic analysis using mass spectrometry identified adenyl cyclase-associated protein 1 (CAP1) as a promising potential biomarker. The verification cohort consisted of 14 HC, 15 MCI patients, and 10 AD patients, with a total of 39 tear samples. A self-assembled nanoparticle-

mediated fluorescence immunoassay (SNAFIA) was applied to detect the target protein, CAP1, in human tear fluid samples. The presence of the CAP1 protein generated a sandwich immunocomplex with an antibody-conjugated magnetic nanoparticle (Ab-MNP) and an antibody-conjugated polymeric nanoprobe (Ab-PNP), enabling the discrimination of the three groups of participants based on the analysis of the fluorescence signals from the SNAFIA assay. The schematic was created with BioRender.com.

Revised Supplementary Table 5:

Parameter	HC	MCI	AD
N	7	7	7
Age (years)	68.7 ± 6.4	71.1 ± 7.7	73.3 ± 5.8
Sex (M : F)	5 : 2	5 : 2	5 : 2
Amyloid-β positivity (Aβ- : Aβ+)	7 : 0	0 : 7	0 : 7
MMSE	29.2 ± 1.2	24.2 ± 1.9 ^a	16.4 ± 4.2 ^{a, b}

Supplementary Table 1 Demographics of participants of the discovery cohort for tear protein biomarker profiling in this study. Statistical analysis was performed by multiple comparisons of Brown-Forsythe and Welch one-way analysis of variance tests (^a $p < 0.001$ for the comparisons between healthy control (HC) and each group. ^b $p < 0.01$ for the comparisons between mild cognitive impairment (MCI) and Alzheimer's disease (AD) groups). Data present mean ± s.d. ($N = 7$ for each group).

Revised Supplementary Table 9:

Parameter	HC	MCI	AD
N	14	15	10
Age (years)	66.9 ± 8.5	74.9 ± 7.9	76.9 ± 8.8
Sex (M:F)	8 : 6	9 : 6	3 : 7
Amyloid-β positivity (Aβ- : Aβ+)	8 : 4	4 : 8	5 : 5
MMSE	28.0 ± 2.1	26.1 ± 2.9	16.1 ± 5.4 ^{a, b}

Supplementary Table 10 Demographic characteristics of participants of verification cohort for clinical diagnostics application of SNAFIA. Statistical analysis was performed by multiple comparisons of Brown-Forsythe and Welch one-way analysis of variance tests (^a $p < 0.001$ for the comparisons between HC and AD groups. ^b $p < 0.001$ for the comparisons between MCI and AD groups). Data present mean ± s.d. ($N = 14, 15,$ and 10 for each group).

Q2. please include how the tear proteins were retrieved from the thread

Thank you for your comment. We would like to clarify that we did not use threads or Schirmer strips for tear collection in this study. Instead, we employed polyester fiber rods^{1, 2}, as detailed in the *Methods* section of the manuscript. As in the previous study, absorbed tear fluid was recovered from the polyester fiber by batch centrifugation for each subject at 4,400 rpm for seven minutes.

In addition, the decision to use polyester fiber rods in our experiments was based on several considerations. First, the polyester fiber rods absorb tears efficiently and consistently allowing us to collect a sufficient volume of tears for subsequent proteomic and SNAFIA analysis. Second, the rapid collection process using polyester fiber rods reduces the time that the sample is exposed to the surrounding environment, minimizing the risk of protein degradation and external contamination.

Additionally, as discussed in previous literature^{3, 4, 5} different tear collection methods can influence the composition of the collected tear fluids. For example, the use of threads or Schirmer strips can introduce variability in tear protein due to differences in migration rates caused by the absorption properties of the proteins and ocular surface irritation. In contrast, using polyester fiber rods has the advantage of avoiding direct contact with ocular surface cells, eliminating the risk of reflex tear secretion and contamination by surrounding tissue, and quantifying tear volume with fewer pre-analytical steps.

Therefore, we hope that this explanation sufficiently provides the rationale for addressing polyester fiber rods for the collection of tear fluids, and provides full experiment steps we took to obtain the tear proteins. Based on the reviewer's feedback, we have revised the *Methods* section in the manuscript to provide sufficient experimental steps.

Revised text in line 620 of page 30: It was then placed in a 1.5 mL Eppendorf (EP) tube and absorbed tear fluid was recovered from the fiber by batch centrifugation for each subject at 4,400 rpm for seven minutes using an EP centrifuge (Westbury, NY). The retrieved tear proteins were stored at –70 °C until a mass spectrophotometric assay was performed.

Q3. the number of samples (39) included into the analysis is low and for verification this is too low number

Thank you for raising the concerns regarding the sample size. First, our study was initiated with a sizable enrollment of participants, including 80 HC, 110 MCI, and 112 AD individuals. All subjects received ¹⁸F-florbetaben positron emission tomography (PET) imaging for amyloid- β measurement. This participant enrollment demonstrates our commitment to ensuring a thorough and representative sample. As with many biomedical studies, we experienced sample losses due to sample quality, quantity, and other unforeseen circumstances. Nevertheless, our samples were representative and yielded insightful results. It is also important to note that samples from the initial cohort were used in another concurrent study, which used samples from 44 HC, 47 MCI, and 56 AD. As a result, the remaining tear samples, i.e., tear samples from 21 HC, 22 MCI, and 17 AD, were used in the present study (**Fig. 1**). While this sample allocation may have affected the number of samples required for both studies, we believe that by separating the samples required for each stage, we can ensure the reliability and meet the integrity of both discovery and validation cohort studies.

In proteomic studies, especially those focused on biomarker discovery and verification, the depth and quality of analysis often outweigh the importance of the sheer number of samples. Our study was systemically structured with a two-pronged approach, covering both discovery and verification stages, and this organization inherently strengthens the credibility of our findings. The discovery phase focused on identifying potential tear biomarkers for the diagnosis of AD, while the verification phase confirmed the preliminary performance results of the identified biomarkers. Thus, the consistent results between two cohorts for one candidate biomarker emphasize the reliability of the results, even if the overall

sample size is limited.

Another challenge is that the process of collecting tear samples from elderly subjects is complex, requires specialized skills, and often ends up with minimal volumes. Given that this study focuses on AD, which predominantly affects people over the age of 65, and the average age of the actual recruited subjects was 60–80, we would like to acknowledge that there were challenges in collecting consistent and sufficient volumes of tears, such as reduced tear production in the elderly or potential discomfort during collection.

Furthermore, the cohorts used in our study were carefully selected to ensure the representativeness of the findings, giving our results external validity. Biological samples are inherently variable, and even in validation experiments with large sample sizes, the presence of confounding variables can mask true discoveries. Studies with similar sample sizes are not common in the field of proteomics and biomarker research, especially if they target specialized populations or use intricate collection methodologies^{6, 7, 8, 9}. Therefore, we believe that methodological precision and consistency of results can offset the limitations of sample size.

Based on these scientific considerations, we are confident in the robustness and relevance of our findings. We also believe that our sample size is methodologically justified and aligns with the objectives of our study.

Q4. the biomarker discovery and the verification of selected potential markers was done on the same samples decreasing the power of the results

Thank you for raising this point. We apologize for any confusion our manuscript may have caused regarding the use of samples for discovery and verification. To clarify, our research was meticulously designed to incorporate two separate human cohorts: one for the discovery phase and the other for the verification phase. This approach was adopted not only to enhance the robustness of our findings but also to rigorously validate the obtained results. By utilizing separate cohorts for each phase to identify potential biomarkers and then independently validate them, we strengthened the credibility and reliability of our findings.

Therefore, we have made this distinction more evident in the revised manuscript to prevent any further misunderstandings.

New text in line 106 of page 5: In this study, a total of 60 participants are used and organized into two distinct cohorts: discovery and verification (**Fig. 1**). The discovery cohort consists of a total of 21 samples and identifies a potential biomarker, adenylyl cyclase-associated protein 1 (CAP1), through proteomic experiments. The verification cohort consists of 39 samples in total, and the CAP1 protein identified in the discovery phase is applied to the SNAFIA immunoassay proposed in this study.

Q5. it is not clear why APOE was selected for study

Thank you for your comment regarding the selection of apolipoprotein E (APOE) for the study. We are pleased to take this opportunity to elucidate the rationales for selecting APOE. The primary aim of our study was to demonstrate the efficacy of the SNAFIA test, a novel immunodiagnostic platform designed for detecting protein biomarkers in tears. Given the innovative nature of the SNAFIA test, it was crucial to first ascertain the testing reliability using a well-established biomarker.

APOE, especially the APOE4 allele, stands for the most important genetic risk factor for AD, and its association with an elevated risk of developing AD has been well-documented in numerous studies^{10, 11, 12, 13}. It is also present in both the central nervous system and peripheral biofluids, making it an excellent candidate for validation purposes^{14, 15, 16}. We chose APOE as a biomarker for two main purposes. First, utilizing APOE, with its well-known association with AD, provided us with a robust benchmark against which to measure the sensitivity and specificity of the SNAFIA test. We believe this validation step is essential to instill confidence in the newly proposed methodology in the scientific community. Second, we demonstrated the adaptability of the SNAFIA test to different biofluids in that we were able to detect apolipoproteins in serum and tears. Interestingly, our proteomic experiment did not reveal significant expression of APOE in the tears of MCI or AD patients compared to the HC groups; however, this underscores our overarching goal of exploring the potential of tear-based biomarkers for AD, a relatively nascent area of research.

In essence, the strategic choice of APOE was to validate the capabilities of the SNAFIA test using a recognized AD biomarker, and subsequent exploration of CAP1 and other potential biomarkers was based on this foundational validation.

Therefore, we hope that this explanation sufficiently explains the rationale for selecting APOE for this study. Based on the reviewer's feedback, we have revised the manuscript to provide a clearer explanation of the rationale for including APOE biomarkers in the SNAFIA trial study.

New text in line 420 of page 22: As depicted in **Supplementary Fig. 13a**, to establish the versatility of the SNAFIA assay for a variety of human biofluids, including tear fluid and serum, we conducted spiking experiments involving the introduction of APOE protein into human serum.

Q6. very few references are used in the discussion part

We thank the reviewer for this comment. In the revised manuscript, we have added references in the *Discussion* section to support our claims and provide a more comprehensive context.

New reference in the *Discussion* section:

[1] Serrano-Pozo A., Das S. & Hyman B. T. APOE and Alzheimer's disease: advances in genetics,

pathophysiology, and therapeutic approaches. *Lancet Neurol* **20**, 68-80 (2021).

[37] Schneider F. et al. Mutual functional dependence of cyclase-associated protein 1 (CAP1) and cofilin1 in neuronal actin dynamics and growth cone function. *Prog Neurobiol* **202**, 102050 (2021).

[39] Kakurina G. et al. A pilot study of the relative number of circulating tumor cells and leukocytes containing actin-binding proteins in head and neck cancer patients. *J Biomed Res* **37**, 213-224 (2022).

[41] Rust M. B. & Marcello E. Disease association of cyclase-associated protein (CAP): Lessons from gene-targeted mice and human genetic studies. *Eur J Cell Biol* **101**, 151207 (2022).

[46] Huang J. et al. Blood levels of MCP-1 modulate the genetic risks of Alzheimer's disease mediated by HLA-DRB1 and APOE for Alzheimer's disease. *Alzheimers Dement* **19**, 1925-1937 (2023).

[47] Sun R. & Xie C. Peripheral ApoE4 Leads to Cerebrovascular Dysfunction and Abeta Deposition in Alzheimer's Disease. *Neurosci Bull* **39**, 1330-1332 (2023).

Points to be addressed

Q7. According to the current knowledge not APOE itself, but APOE e4 allele is a risk factor for AD. Please correct this in the text.

We thank the reviewer for highlighting the difference between the general APOE protein and its specific allele APOE e4, which is known to be a prominent genetic risk factor for AD. The overarching objective of this study was to explore the potential of human tears as a novel biofluid for biomarker discovery and to validate the efficacy of our newly developed SNAFIA test. In this context, we utilized APOE, which has a well-established association with AD, as a representative protein without distinguishing between specific alleles. This was a methodological choice to demonstrate the broad detection capabilities of the SNAFIA test rather than its accuracy in identifying particular APOE subtypes.

However, we recognize the importance of specificity in biomarker detection, especially for genetic risk factors such as APOE e4. As part of our ongoing research efforts, we are considering studies that delve deeper into the detection capabilities of the SNAFIA test, particularly by targeting different alleles of APOE. Such studies will not only enhance the specificity of our test but also contribute to a broader understanding of the role of APOE alleles in various biofluids.

We thank the reviewer again for their insightful feedback. We hope the revised manuscript clarifies our research objectives and emphasizes our current findings and future directions.

Revised text in line 271 of page 14: APOE is one of the components of lipoproteins that transport and metabolize lipids, and polymorphisms in the APOE gene, especially the APOE e4 allele, are known to profoundly affect the risk of sporadic AD⁴¹.

Q8. Please give information on which APOE is recognized by the antibody. APOE4 or all APOEs?

Thank you for the comment. As mentioned in our response to the reviewer's question #7, the primary goal of this study was to validate the efficacy of the SNAFIA test in detecting protein biomarkers in human biofluids, particularly tears. Consistent with this goal, we used antibody pairs that recognize APOE in general rather than specific subtypes such as APOE4. We focused on the broad detection of potential biomarkers in tears rather than the differentiation of specific APOE alleles, which is already well-established in other biofluids, including blood and CSF.

However, we acknowledge the significance of differentiating between APOE alleles, especially given the strong association of the APOE4 allele with AD risk. In future studies, we aim to enhance the detection capabilities of the SNAFIA test to discern specific subtypes of APOE in tear samples, providing even more vital insights into the association between APOE and AD. We have, therefore, provided a clear description of the specificity of the antibodies used in the revised manuscript.

Revised text in line 274 of page 14: To evaluate the utility of the SNAFIA in human serum, Ab-MNPs and Ab-PNPs were coated with IgG antibodies that recognize all APOE proteins, not just the APOE4 subtype.

Q9. Sentence in line 299 should be modified. There is no presented evidence that "SNAFIA has the ability to directly detect AD biomarkers as low as 0.06 fM". The data refer only to APOE.

We appreciate the reviewer's comments on the details. As the reviewer's comment, we have revised the sentence in line 299 to state that we demonstrated that the SNAFIA test can detect APOE at concentrations as low as 0.06 fM. We apologize for any confusion caused by the original sentence and have ensured that our claims are accurately supported by the data presented.

Revised text in line 299 of page 15: Thus, SNAFIA has the ability to directly detect APOE at concentrations as low as 0.06 fM, even in complex fluids containing multiple interfering substances, such as serum.

Q10. Please show results for APOE spiked in tears. What is the dilution factor for tears?

As suggested by the reviewer, we further evaluated the sensitivity of the SNAFIA test by spiking different concentrations of APOE in tears. Here, we used commercially available artificial tear fluid (ATF) without dilution. We then diluted APOE protein at a 5-fold volume ratio to prepare a final concentration of 0.0128–5000 fM in ATF solution and used the SNAFIA test to determine the detectable concentration range and limit of detection (LOD). As a result, we observed an APOE concentration-dependent signal change in ATF, similar to the analysis of PBS and serum samples, and the calculated LOD was 0.0728

fM (Fig. R1 and Revised Supplementary Table 3). This result shows that the SNAFIA-based assay performs well in detecting APOE in biofluids such as serum and ATF without inducing significant signal attenuation.

New Figure R1:

Supplementary Fig. 17 Sensitivity test of SNAFIA with increasing concentration of APOE protein. (a) Normalized FI of SNAFIA with APOE protein spiked into PBS, 10% (v/v) human serum, and ATF (excitation at 475 nm, emission at 500 nm). (b) Normalized FI of STNAFIA with APOE protein spiked in ATF. The LOD of SNAFIA for PBS, serum, and ATF samples was determined by 3 σ and reported to be 0.0542, 0.0586, and 0.0728 fM, respectively. The measurement was performed in triplicate, and all reported values represent mean \pm s.d.; $n = 3$ for independently repeated tests.

Revised Supplementary Table 3:

Parameter	PBS	Human serum	ATF
Slope	0.285	0.265	0.280
Y-intercept	0.367	0.357	0.357
X-intercept	-1.29	-1.35	-1.28
Equation	$Y = 0.285 \cdot X + 0.367$	$Y = 0.265 \cdot X + 0.357$	$Y = 0.280 \cdot X + 0.357$
R squared	0.977	0.949	0.953
Limit of detection (fM)	0.0542	0.0586	0.0728
Limit of quantification (fM)	0.0609	0.111	0.154

Supplementary Table 8 Comparison of the analytical sensitivity of the SNAFIA test targeting apolipoprotein E (APOE) protein in PBS, human serum, and ATF using analytical constants of the linear regression curve. The LOD and LOQ were based on the 3 σ and 10 σ methods, respectively.

Q11. Please show results for the concentration measurement of APOE in tear and serum of controls and/or patients. I think it is important to demonstrate that the test performs well with real biological fluids.

We appreciate the reviewer's important comments. We agree with the reviewer that it is essential to test the concentration of APOE in tear and serum samples from human participants to demonstrate that the SNAFIA test works well in real biological samples. However, we would like to clarify that our primary interest is to utilize the SNAFIA-based protein detection system to validate the clinical diagnostic value of tear-based AD biomarkers, a relatively unexplored area of research. In addition, as mentioned in our response to the reviewer's question #5, we would like to reiterate that our goal in this

study is to validate the efficacy of the SNAFIA methodology in detecting protein biomarkers in human biofluids, with a particular focus on tears. To this end, our study was based on a workflow that involved screening novel AD tear markers by proteomic analysis and validating the ability of the immunodiagnostic SNAFIA platform to diagnose AD patients and distinguish disease progression.

Therefore, prior to proteomic analysis, we first validate the detection performance of our designed SNAFIA using APOE protein, a well-recognized AD marker. Subsequently, we selected a candidate AD tear marker (e.g., CAP1) and clinically validated it using tears from human participants. We would also like to clarify that our attempt to detect APOE protein in serum samples was to explore the applicability of SNAFIA in various biofluids and that detecting AD tear markers in serum samples derived from patients and healthy controls was not the main objective of our study. We hope that our responses provide sufficient answers to the reviewer's questions.

Q12. I think the authors do not have enough evidence to make the following statement: "This demonstrates that the SNAFIA platform has excellent selectivity, even in complex samples with non-specific proteins." Having combined two proteins is very far from the complex biological samples. Please modify the statement accordingly.

We have amended the sentence as follows to reflect the reviewer's comments. We further performed a selectivity test of SNAFIA for CAP1 as the target protein in samples mixed with the non-target proteins APOE and ACHE. Similar to the results of the selectivity test for the APOE target protein performed earlier, the SNAFIA platform was able to effectively determine the presence of the target protein in samples mixed with two non-target proteins (Fig. R2). This newly obtained result has been added to the revised Fig. 4 in the manuscript.

Revised text in line 348 of page 17: This demonstrates that the SNAFIA exhibits distinct target detection performance within specimens containing a mixture of both target (CAP1 protein) and non-target proteins (APOE and ACHE proteins), with a p -value less than 0.0001.

Fig. 4 (d) Selectivity of SNAFIA for various AD biomarker candidates and their mixtures in PBS. The

concentrations of CAP1, apolipoprotein E (APOE), and acetylcholinesterase (ACHE) were varied to 1 pM (1×), 5 pM (5×), and 10 pM (10×). Statistical analysis was performed by multiple comparisons of Brown-Forsythe and Welch one-way analysis of variance tests ($****p < 0.0001$). The measurement was performed in triplicate, and all reported values represent mean \pm s.d.; $n = 3$ repeated tests.

Q13. It is not clear why APOE was tested in serum and PBS and CAP1 in artificial tears? Please perform experiments using both SNAFIA for APOE and CAP1 for PBS, serum, artificial tear and pooled tear from patients. Please show the results of spiked proteins into these matrixes and the amount quantified in the serum and tear.

Thank you for raising this point. We recognize the importance of providing a clear rationale for the selection of sample groups in an experimental design and appreciate the opportunity to elucidate this. The decision to test for APOE in serum and PBS was a logical choice for initial validation of the SNAFIA test because the presence of APOE in serum is well established. PBS, a physiological solution, was used to serve as a simple and basic matrix to assess the performance of the test. On the other hand, CAP1 was tested in artificial tears because the primary interest was to explore the potential of tear-based biomarkers for AD. Since tears were the main focus of the study, it was essential to validate the efficacy of the SNAFIA test in a matrix that closely resembles the natural tear environment.

However, we agree with the reviewer's suggestion that the detection performance of SNAFIA for APOE and CAP1 should be evaluated in a variety of matrices, including PBS, serum, and artificial tears. In response to the reviewer's question #10, we have investigated the detection sensitivity of APOE protein by spiking in three different matrices and present the obtained results (**Fig. R1**). Similarly, we obtained the detection sensitivity and LOD of CAP1 protein by SNAFIA with three different matrices at known concentrations (**Fig. R3**).

While this approach may indeed provide a broader validation spectrum, we emphasize again that the current study was designed with the specific objective of detecting tear AD markers in mind, and the matrices we chose were most relevant to the research goals we set to validate the performance of the SNAFIA test for APOE and CAP1 markers, respectively.

New Figure R3:

Supplementary Fig. 16 Sensitivity test of SNAFIA with increasing concentration of CAP1 protein. (a) Normalized FI of SNAFIA with CAP1 protein spiked into PBS, ATF, and 10% (v/v) human serum (excitation at 475 nm, emission at 500 nm). (b) Normalized FI of SNAFIA with CAP1 protein spiked in 10% (v/v) serum. The LOD of SNAFIA for PBS, serum, and ATF samples was determined by 3σ and reported to be 0.282, 0.236, and 0.451 fM, respectively. The measurement was performed in triplicate, and all reported values represent mean \pm s.d.; $n = 3$ repeated tests

Revised Supplementary Table 7:

Parameter	PBS	ATF	Human serum
Slope	0.271	0.278	0.273
Y-intercept	0.172	0.204	0.165
X-intercept	-0.635	-0.728	-0.603
Equation	$Y = 0.271 * X + 0.172$	$Y = 0.278 * X + 0.204$	$Y = 0.273 * X + 0.165$
R squared	0.975	0.944	0.968
Limit of detection (fM)	0.283	0.236	0.451
Limit of quantification (fM)	0.447	0.406	0.823

Supplementary Table 6 Comparison of the analytical sensitivity of the SNAFIA test targeting cyclase-associated protein 1 (CAP1) protein in PBS, artificial tear fluid (ATF), and human serum solutions using analytical constants of the linear regression curve. The limit of detection (LOD) and limit of quantification (LOQ) were based on the three-sigma (3σ) and ten-sigma (10σ) methods, respectively.

Q14. Based on which data the authors define progressively up and gradually downregulated proteins?

The reviewer raised an important point. In our study, the differential expression of 75 tear proteins was first determined by comparing the relative expression levels in patients with MCI or AD to the HC group (MCI/HC or AD/HC). For this determination, we set specific thresholds, i.e., proteins exhibiting a fold change greater than 1.5 ($p < 0.01$) were classified as upregulated (64 tear proteins), and proteins showing a fold change less than 0.67 ($p < 0.01$) were considered downregulated (11 tear proteins) (Fig. R4).

Each upregulated or downregulated tear protein was further categorized as follows: upregulated proteins with a trend toward a significant increase in the fold change of each protein expression in AD/HC compared to MCI/HC (defined as 'progressively upregulated tear proteins') and down-regulated proteins with a trend toward a significant decrease in the fold change of each protein expression in AD/HD compared to MCI/HC (defined as 'gradually down-regulated tear proteins').

We realized that the definitions of progressively upregulated and downregulated tear proteins were not fully described in the original manuscript, and therefore, we added the following sentences to the revised manuscript. We also added a part 'Quantitative global proteome profiling and data processing for human tear biomarker discovery' to the *Method* section to describe the quantification methods of the proteomic analysis results.

New text in line 183 of page 10: 'Progressively up-regulated' means an increasing trend in fold

change from the MCI/HC comparison to the AD/HC comparison. This means that the protein's expression level increased more prominently when comparing AD patients to HC than when comparing MCI patients to HC. 'Gradually down-regulated' means a decreasing trend in fold change from the MCI/HC comparison to the AD/HC comparison. This indicates that the protein's expression level decreased more significantly when comparing AD patients to HC than when comparing MCI patients to HC (Fig. 2b and Supplementary Table 2).

New Figure R4:

Fig. 2 (b) Heatmap of differentially expressed proteins (DEPs) determined by comparing the relative expression levels in patients with MCI or AD to the HC group (MCI/HC or AD/HC). Proteins exhibiting fold changes greater than 1.5 classified as upregulated (left) and proteins showing a fold change less than 0.67 classified as downregulated (right). Statistical analysis was performed for each protein using a *t*-test ($p < 0.01$).

Q15. Please also comment in the text the components of chemical barrier having role in defense as shown on figure 4 b. There are available information on the changes of the components of

the chemical barrier and the network of antimicrobial and immunomodulatory peptides in tear and in AD in the literature. I think it is important to highlight besides the other functions these functions as well.

Thank you for highlighting the significance of discussing the broader roles of tear proteins, particularly those associated with chemical barrier, antimicrobial, and immunomodulatory functions. While our study primarily centered on the novel AD-associated protein CAP1, we recognize the importance of providing a holistic view of the tear fluid environment in AD.

Although we focused mainly on CAP1, we recognized the importance of discussing these broader changes to provide readers with a comprehensive understanding of the tear fluid alterations in AD. In the revised manuscript, we discussed these proteins and their potential roles in the context of tear fluid defense in more detail based on the available literature. We believe this will provide a more balanced perspective on the changes in tear fluid composition in AD beyond our focus on CAP1.

New text in line 209 of page 11: In **Fig. 2b and 2c**, while our study primarily focused on the novel protein CAP1, several proteins with roles in defense mechanisms and the chemical barrier are also depicted. For instance, proteins like LCP1 (Lymphocyte Cytosolic Protein 1), CORO1A (Coronin 1A), HMGB1 (High Mobility Group Box 1), RETN (Resistin), NAMPT (Nicotinamide Phosphoribosyl transferase), and HSPA5, HSPB1 (Heat Shock Proteins) have known roles in immune responses, while others like SCGB1D1 and SCGB2A1 (Secretoglobins) possess immunomodulatory properties. Additionally, proteins such as LACRT (Lacritin) are specific to the eye and influence tear composition and defense capabilities of tears. LCN1 (Lipocalin-1) might play a role in modulating the chemical environment of tears, indirectly contributing to its defense mechanisms. Recent literature has indeed highlighted changes in the components of the chemical barrier and the network of antimicrobial and immunomodulatory peptides in tear fluid in the context of AD¹⁷. The altered composition of the chemical barrier, along with the reduced level of defense proteins, might imply an increased risk of ocular infections in AD patients. However, there hasn't been a reported increase in ocular infections in AD patients in the scientific literature.

Q16. Why results with APOE are shown if only CAP1 SNAFIA was used later? It is not clear why APOE was included if CAP1 is the potential biomarker.

Thank you for raising this point. As detailed in our response to the reviewer's questions #5 and #13, the primary objective of our study was to validate the efficacy of the newly developed SNAFIA test for the rapid, simple, and sensitive detection of AD-associated tear proteins as a diagnostic tool for AD biomarkers. Before using a potential AD tear protein marker such as CAP1, it was crucial to obtain a database of the sensitivity and selectivity of SNAFIA using a well-known AD biomarker. APOE, especially the APOE4 allele, is a well-established genetic risk factor for AD. Since numerous studies have consistently shown a high association with AD, we considered APOE to be an ideal candidate marker for initial validation. More importantly, we wanted to gain confidence in the system by

demonstrating the ability of the SNAFIA test to selectively and sensitively detect APOE in various biofluids.

After successfully validating the SNAFIA test using APOE, we proceeded to explore other potential tear-related biomarkers and eventually targeted the CAP1 protein, which was identified as a promising candidate in proteomic analysis of tear fluid from AD patients, to assess its value as a tool for AD diagnosis and progression prediction. While we sought to demonstrate the versatility and reliability of the SNAFIA test by including results obtained with APOE, a universal AD marker, we would like to clarify that CAP1 is nevertheless the tear protein marker for AD diagnosis that is the focus of this study.

Therefore, to avoid this confusion, we have made a more apparent distinction in the revised manuscript between the validation phase of the SNAFIA system using APOE and the subsequent exploration of CAP1 as a potential AD biomarker.

New text in line 420 of page 22: As depicted in **Supplementary Fig. 13a**, to establish the versatility of the SNAFIA assay for a variety of human biofluids, including tear fluid and serum, we conducted spiking experiments involving the introduction of APOE protein into human serum.

Q17. According to the presented data CAP1 does not fulfill all the requirements for a biomarker, please refer to it in the text as potential biomarker instead of biomarker. Only after validation on large patient number (thousands of patients) we can call a protein biomarker. The use of “biomarker” term in the text should be revisited and used accordingly.

We appreciate the reviewer's thoughtful feedback. As the reviewer points out, we recognize that the term “biomarker” carries certain connotations, particularly concerning validation in large cohorts. We intended to propose CAP1 as a promising candidate for the diagnosis of AD in tears based on our findings, and we recognize that rigorous validation in large patient cohorts is required before we can confirm this protein as an AD biomarker. Therefore, we have revisited the manuscript in light of the reviewer's comments and decided to consistently refer to CAP1 as a “**potential biomarker**” or “**biomarker candidate**” in the revised manuscript until further validation studies are conducted in larger patient populations.

New text in line 39 of page 2: Using a discovery cohort of 21 subjects, proteomic analysis identified adenylyl cyclase-associated protein 1 (CAP1) as a potential tear biomarker.

Revised text in line 105 of page 5: for the highly sensitive detection of **biomarker candidates** in AD clinical tear samples.

Revised text in line 108 of page 5: (Ab-MNP-**CAP1**-Ab-PNP)

Revised text in line 117 of page 5: concentration of the AD biomarker candidate in tear fluid

Q18. In the sentence “SNAFIA test was able to accurately identify the MCI and AD patient groups in the HC participants” the preposition in is inappropriate. Please change accordingly.

We thank the reviewer for this observation. We have made a change in this sentence accordingly.

Revised text in line 458 of page 22: Thus, the SNAFIA test was able to accurately identify the MCI and AD patient groups from the HC participants with a sensitivity of 73.3% and specificity of 100%, and a sensitivity of 90% and specificity of 100%, respectively.

Q19. Please change the scales to log scale on Suppl. Fig. 11. and 15

As requested by the reviewer, we converted the linear scale in **Supplementary Fig. 11 and 15** to a logarithmic scale. We found that the modified graphs fit the exponential regression curve better, and the coefficients of determination (R-squared values) were slightly improved. We also revised the figure captions accordingly.

Revised Supplementary Fig. 11:

Supplementary Fig. 16 ELISA and hSNAFIA tests with different concentrations of APOE protein ranging from 0 to 2000 nM. APOE dose-responsive curves and exponential regression obtained by (a) ELISA and (b) hSNAFIA tests, respectively. Data present mean \pm s.d. for three independent experiments ($n = 3$).

Revised Supplementary Fig. 15:

Supplementary Fig. 12 ELISA and hSNAFIA tests with different concentrations of CAP1 protein ranging from 0 to 25000 nM. CAP1 dose-responsive curves and exponential regression obtained by (a) ELISA and (b) hSNAFIA tests, respectively. Data present mean \pm s.d. for three independent experiments ($n = 3$).

Q20. Please clarify what is the difference between the patients shown in Suppl. Table 5 and 9. In the methods where the tear collection is described Suppl. Table 5 is mentioned, while in the text Suppl. Table 9.

We thank the reviewer for pointing out the inconsistency in the references to **Supplementary Tables 5 and 9** in the manuscript. To clarify, **Supplementary Table 5** presents the demographic characteristics of participants in the **discovery cohort** used for tear protein biomarker profiling, and **Supplementary Table 9** provides the demographic characteristics of participants in the verification cohort for the clinical diagnostic application of SNAFIA.

Although we employed the same tear collection method in both cohorts to maintain uniformity of sample collection at different stages of the study, we acknowledge that we made an unintentional mistake in referencing a table in the *Methods* sections. In the main text describing tear samples for proteomic analysis and SNAFIA testing, we correctly referenced **Supplementary Table 5** and **Supplementary Table 9**, respectively.

To rectify this, we have clearly and consistently referenced the appropriate tables in the relevant text of the revised manuscript. We have also included a sentence highlighting the consistent tear collection methods used in the discovery and validation cohorts to eliminate ambiguity.

Revised text in line 615 of page 30: Detailed demographics are summarized in **Supplementary Tables 1 and 10**.

Revised text in line 616 of page 30: To perform global proteome profiling for the **discovery phase** and **SNAFIA assay for the verification phase**, tear fluid was collected from participants using a bonded 2.0 \times 10 mm polyester fiber rod (TRANSORB® WICKS, FILTRONA, VA, USA).

New text in line 596 of page 30: The study was designed to incorporate two distinct human

cohorts: a discovery cohort comprised of 7 healthy control (HC), 7 MCI, and 7 AD participants; and a verification cohort consisting of 14 HC, 15 MCI, and 10 AD participants.

Q21. Please give a detailed description on how the proteomics analysis was carried out. A detailed description of the sample preparation and analysis: digestion, more details on TMT labeling, sample cleanup, chromatographic separation, fractionation, mass spec conditions (settings), type of data acquisition method applied is required.

At the request of the reviewer, we have detailed the entire process of proteomics analysis below, breaking down each step.

1) Human tear sampling

To collect the tear fluid from participants, a 2.0 × 10 mm polyester fiber rod (TRANSORB® WICKS, FILTRONA, Richmond, VA, USA) was used as previously reported¹. The polyester wick was applied to the tear meniscus at the margin of the patient's lower eyelid. The polyester fiber rod was then removed, placed into a 1.5 mL Eppendorf tube, and absorbed tear fluid was recovered from the fiber by batch centrifugation for each subject at 4,400 rpm for seven minutes using an EP centrifuge. The retrieved tear proteins were stored at -70 °C until mass spectrophotometric analysis was performed. Total tear protein concentration in each sample was determined using a Pierce™ Micro BCA Protein Assay Kit (Thermo Fisher Scientific).

2) In-solution digestion

Proteins (100 µg) of the pooled tear fluids from each group were digested into peptides by in-solution digestion, as shown in **Fig. 2a**. In detail, 8 M urea in 100 mM ammonium bicarbonate (Sigma) was mixed with each fluid sample in a 1:1 volume ratio, and the mixture was incubated at room temperature (RT) for 30 min. The samples were then reduced with 10 mM dithiothreitol at RT for 30 min, followed by alkylation with 30 mM iodoacetamide for 45 min in the dark. Trypsin was added to the samples at a volume ratio of 1:50 and incubated at 37 °C overnight. The trypsin reaction was quenched with 0.8% trifluoroacetate, and the resulting peptides were desalted on a C18 Harvard macro spin column (Harvard Apparatus, Holliston, MA, USA). The obtained peptides were dried in a Speed-Vac concentrator and stored at -80 °C.

3) TMT labeling

Each peptide sample was labeled with tandem mass tags (TMT), and isobaric mass tagging reagents, according to the manufacturer's instructions (Thermo Fisher Scientific). For each sample of 50 µg peptides, 100 µL of 100 mM triethylammonium bicarbonate (TEAB) were added. TMT reagents were resuspended in anhydrous acetonitrile (ACN) and then added to each sample (126: HC_Replicate 1, 127: MCI_Replicate 1, 128: AD_Replicate 1, 129: HC_Replicate 2, 130: MCI_Replicate 2, 131: AD_Replicate 2). After 1 h, the reaction was quenched with 5% hydroxylamine at RT for 15 min. The TMT-tagged samples were collected in a tube, concentrated by high-speed vacuum centrifugation, and

then high-pH fractionated.

4) High-pH reverse-phase liquid chromatography fractionation

Peptide separation was performed using high-pH reverse-phase liquid chromatography fractionation based on peptide hydrophobicity. Agilent 1260 Series HPLC System (Agilent Technologies, CA, USA) was used to divide the sample into 12 fractions. An Accuser 150 C18 LC column (150 mm × 2.1 mm, 4 μm, Thermo Fisher Scientific) was used for fractionation with the high-pH buffer A and B; 10 mM ammonium formate (pH 10) as mobile phase A and 10 mM ammonium formate in 90% ACN (pH 10) as mobile phase B. The gradient was as follows: 0–10 min, 5% B; 10–70 min, 35% B; 70–75 min, 70% B; 75–85 min, 70% B; 85–90 min, 5% B; and 90–105 min, 5% B. A total of 96 fractions were collected in a 96-well plate every 1 min from 10 min to 105 min of gradient time. The 96 fractions were non-contiguously concatenated into 12 fractions by concatenating one fraction from each section (*i.e.*, Fraction 1: #1-#13-#25-#37-#49-#61-#73-#85, Fraction 2: #2-#14-#26-#38-#50-#62-#74-#86, ..., Fraction 12: #12-#24-#36-#48-#60-#72-#84-#96). The separated peptides were collected and dried in a vacuum centrifuge. Each fraction was desalted on a C18 spin column (Thermo Fisher Scientific).

5) Global profiling using LC-MS/MS

Peptides fractionated into 12 fractions were resuspended in 0.1% formic acid (FA) in deionized water and analyzed using a Q Exactive orbitrap hybrid mass spectrometer (Thermo Fisher Scientific) coupled with a nanoACQUITY UPLC system (Waters, MA, USA). For proteome profiling analysis, the 180 min gradient method was used at a flow rate of 300 nL/min: from 5% to 40% of solvent B for 130 min, from 40% to 80% of solvent B for 5 min, holding at 80% of solvent B for 10 min, and equilibrating the column at 1% of B for 30 min (solvent A: 0.1% FA in deionized water, solvent B: 0.1% FA in ACN). The peptides were eluted through a trap column and ionized through an EASY-spray column (50 cm × 75 μm) packed with 2 μm C18 particles at an electric potential of 1.8 kV. Full mass spectrometry (MS) data were acquired in a scan range of 400–2,000 Th at a resolution of 70,000 at m/z of 200, with an automated gain control (AGC) target value of 1.0×10^6 and a maximum ion injection of 100 ms. The maximal ion injection time for MS/MS was set to 100 ms at a resolution of 17,500. The 10 most abundant ions were separated with an isolation window of ± 0.8 Th with an isolation offset of 0.5 Th and fragmented by higher-energy collisional dissociation (HCD) with a normalized collision energy of 35. The dynamic exclusion time was set to 30 s.

We summarize the above description as 'Proteomic analysis process for human tear biomarker discovery' in the *Methods* section.

Q22. Please describe which software were used and how the mass spectrometry data were processed. Please indicate the database version, applied settings and the FDR correction of the data.

We processed the obtained LC-MS/MS spectra using the Proteome Discoverer software version 2.1 (Thermo Fisher Scientific) and searched against the UniProt human database with the SEQUEST

HT® search engine. Static modifications we applied included carbamidomethylation (C) and TMT six-plex (N-terminal, lysine (K)). In contrast, dynamic modifications encompassed methionine oxidation. The resulting peptide hits were filtered using the Percolator algorithm to achieve a maximum false discovery rate (FDR) of 1%.

For quantification, we calculated reporter ratios with a mass tolerance of ± 10 ppm using the TMT 6-plex method within the Proteome Discoverer software. Protein intensities were then normalized by total sum-scaling to ensure comparability. We then calculated the ratio of mean values of the normalized protein intensities for each protein and determined the *p*-value for each protein using a *t*-test. We defined differentially expressed proteins (DEPs) in MCI and AD patients compared to HC groups based on a fold change greater than 1.5 and less than 0.67 and a *p*-value less than 0.01.

We describe it as ‘Quantitative global proteome profiling and data processing for human tear biomarker discovery’ in the *Methods* section. We hope that our description has provided sufficient explanation for the reviewer’s comment.

Q23. Please show a list of quantified proteins in Supplementary table.

We thank the reviewer for this suggestion. Based on the reviewer’s comment, we have included **Table R1** in the *Supplementary Information* of the revised manuscript, which describes in detail all the proteins quantified in this study.

New Table R1:

No.	DEPs		
	Gene Symbol (MCI_AD_Common_DEP)	Fold change (MCI/HC)	Fold change (AD/HC)
1	RETN	2.34	1.58
2	CST2	2.22	1.86
3	SERPINA2	2.05	1.86
4	AGR2	2.02	1.64
5	BASP1	1.96	1.63
6	CES1	1.87	1.53
7	LRRC59	1.76	1.77
8	ALCAM	1.73	1.60
9	CAP1	1.72	1.86
10	GLUL	1.71	1.69
11	NAMPT	1.70	1.77
12	PDIA3	1.68	1.84
13	HMGB1	1.66	1.82
14	HIST1H1B	1.65	1.56
15	PSMB10	1.65	2.78
16	CAMP	1.65	1.69
17	SUB1	1.64	1.66
18	APOA4	1.63	1.65

19	VAPB	1.61	1.80
20	TOP1	1.61	1.68
21	HMGA1	1.60	1.74
22	FBL	1.60	2.16
23	AARS	1.60	1.67
24	KTN1	1.60	1.78
25	HSPB1	1.59	1.92
26	LMNB2	1.59	1.72
27	DNPEP	1.58	1.81
28	PSMA4	1.57	2.60
29	UTRN	1.57	1.57
30	ARGLU1	1.57	1.68
31	MARCKS	1.57	1.51
32	LCP1	1.57	1.56
33	TMA7	1.57	1.64
34	STK10	1.57	1.54
35	FAHD2A	1.57	1.50
36	ERP29	1.56	1.72
37	MANF	1.55	1.52
38	CCDC50	1.54	1.71
39	TROVE2	1.54	1.96
40	RALY	1.54	1.67
41	UBLCP1	1.54	2.09
42	TXNDC5	1.54	1.53
43	TAF15	1.53	1.66
44	BDH2	1.53	1.61
45	PRPSAP1	1.53	1.65
46	LAGE3	1.53	1.63
47	IGHV3-20	1.53	1.55
48	ILF3	1.53	1.72
49	SERPINC1	1.52	1.59
50	DSG2	1.52	1.61
51	TLN1	1.52	1.63
52	PTBP1	1.52	1.71
53	SNRPA	1.52	1.53
54	IGHG3	1.52	1.57
55	HSPA5	1.52	1.70
56	EEA1	1.51	1.57
57	HMGH4	1.51	1.66
58	STX4	1.51	1.98
59	CORO1A	1.51	1.72
60	SNRNP70	1.51	1.82
61	TCERG1	1.50	1.60
62	ACTB	1.50	1.59
63	PDAP1	1.50	1.50
64	CLINT1	1.50	1.63
65	PLA2G2A	0.67	0.54
66	LCN1	0.67	0.58

67	B2M	0.66	0.65
68	LACRT	0.63	0.52
69	PIP	0.55	0.57
70	PSCA	0.52	0.37
71	PRELP	0.49	0.48
72	SCGB1D1	0.48	0.43
73	SCGB2A1	0.46	0.43
74	IGKV1-9	0.39	0.38
75	SMR3B	0.36	0.49

Supplementary Table 2. Differentially expressed proteins (DEPs) in tear fluid of MCI and AD compared to HC. Sixty-four proteins exhibiting fold changes greater than 1.5 classified as upregulated (red) and eleven proteins showing a fold change less than 0.67 classified as downregulated (blue). Statistical analysis was performed for each protein using a *t*-test ($p < 0.01$).

Q24. Please indicate the statistical analysis used for data evaluation and show the list of DE proteins in a supplementary table and volcano plot in the main text.

We thank the reviewer for the valuable feedback. We used a *t*-test to statistically analyze protein expression levels between each group (MCI, AD, and HC). Also, as we answered the reviewer's question #22, we defined differentially expressed proteins (DEPs) based on proteins with a relative fold change in protein expression greater than 1.5 and less than 0.67 and a *p*-value less than 0.01.

As suggested by the reviewer, we have included **Table R1** in the revised manuscript listing all DEPs identified in the study. In addition, to visualize the relative fold change of DEPs, we have added a heatmap of this result in the main figure to present the fold change data of protein expression observed in MCI and AD patients compared to the HC group. We believe that the MCI/HC comparison and the AD/HC comparison show functional patterns of change for this study, especially since it is important to understand the trends in expression change of tear proteins. Therefore, we believe these additions will improve the clarity and comprehensiveness of our findings.

Revised Figure 2:

Fig. 2 Discovery of novel human tear fluid biomarkers for AD by proteomic analysis. (a) Schematic

representation of the experimental design for global proteome profiling by tandem mass tag labeling using human tear fluid samples from the discovery cohort. (i) Tear fluids were collected non-invasively using a polyester wick and subjected to a proteomics experiment. (ii) Proteins of the pooled tear fluids digested into peptides by in-solution digestion. (iii) Each peptide sample was labeled with tandem mass tags and isobaric mass tagging reagent. Peptide separation was carried out using high-pH reverse-phase liquid chromatography fractionation. (iv) For the proteome profiling analysis, fractionated peptides were analyzed using the Q Exactive orbitrap hybrid mass spectrometer coupled with the nanoACQUITY UPLC system. Full mass spectrometry data were acquired using the Proteome Discoverer software version 2.1. For the quantification, the ratio of mean values of normalized protein intensities was calculated for each protein and a p -value was determined for each protein using a t -test. Heatmap of differentially expressed proteins (DEPs) determined by comparing the relative expression levels in patients with MCI or AD to the HC group (MCI/HC or AD/HC). Proteins exhibiting fold changes greater than 1.5 classified as upregulated (top) and proteins showing a fold change less than 0.67 classified as downregulated (bottom). Statistical analysis was performed for each protein using a t -test ($p < 0.01$). (c) Protein-protein interaction network with significantly enriched biological processes generated from proteins differentially expressed in tear fluid from patients with MCI and AD compared with HC. The inner and outer gray circles represent the relative protein expression (fold change) of each AD and MCI group compared to the HC group, with values close to 1.5 and -1.5 on the log₂ fold change scale shown in red and blue, respectively. The colors of the nodes represent proteins that were significantly increased (red) or decreased (blue) in MCI or AD. Grey lines between nodes indicate biological or physical interactions between proteins. Schematics were created with BioRender.com.

Q25. Please indicate why CAP1 protein was selected for further analyses. This had the highest change compared to controls or based on some other features?

Thank you for inquiring about our choice of the CAP1 protein for further analysis. Our decision to use CAP1 was influenced by several key observations.

First, while CAP1 did not exhibit the most considerable change in expression compared to other proteins, it did show a distinct and consistent trend of progressively increasing expression from HC individuals to MCI patients and further to AD patients (**Fig. R4**). We thought that this apparent pattern of expression over disease progression could be a potential marker for early disease detection, rather than a protein that is simply highly expressed at a particular disease stage. The stepwise increase in CAP1 levels emphasizes its potential as a diagnostic marker, especially in new biosensing platforms that seek to identify early-stage diseases. After discovering the unique presence of CAP1 in the tears of MCI and AD patients, we are planning in-depth studies using several AD mouse models to further understand the role of CAP1 as a potential biomarker in tear fluid.

In essence, CAP1 was selected for its unique expression pattern and its potential diagnostic significance. We anticipate that our future studies will shed more light on its role in AD.

Revised text in line 187 of page 10: Among them, we selected the CAP1 protein, whose relative protein expression was significantly changed in tears from both MCI and AD patients (1.72 and 1.86 compared to HC, respectively) (Fig. 2c).

New text in line 220 of page 11: Overall, it is noteworthy that while the expression levels of CAP1 did not exhibit the most pronounced changes in comparison to other proteins, they did manifest a discernible and consistent trend of incremental expression from HC individuals to those with MCI, and subsequently to AD patients. Such a definitive increasing pattern across disease progression is crucial for early detection rather than a mere high expression at a particular disease stage. The stepwise elevation in CAP1 levels underscores its potential utility as a diagnostic marker, particularly for novel biosensing platforms designed for the identification of diseases in their early stages. The distinctive expression patterns of CAP1 observed in the tear fluid of MCI and AD patients prompted us to plan further experiments including a verification cohort.

Q26. Please upload the proteomics data to public repository, such as PRIDE or ProteomeXchange (<https://www.proteomexchange.org/>) and indicate the accession number in the abstract and in the main text.

Thank you for suggesting that we upload our proteomics data to the public repository. As per the reviewer's comments, we have uploaded the proteomics dataset to the ProteomeXchange Consortium (<https://www.proteomexchange.org/>) via the PRIDE partner repository, and the project accession number is PXD042142. In this regard, we have indicated the project accession number in the *Data availability* section of the revised manuscript.

New text in line 797 of page 39: The mass spectrometry proteomics data have been deposited to the ProteomeXchange Consortium via the PRIDE repository with the dataset identifier (PXD042142).

Q27. Please give details on the network generation. What stringency, value was applied, which String db version was used and please add reference for String DB.

In this study, we aimed to elucidate the biological processes associated with the DEPs in tears linked to MCI and AD. To achieve this, we conducted a gene ontology search focusing on proteins with significance less than 0.01 (p -value < 0.01). For the generation of protein–protein interaction networks, we utilized the STRING database (version 11.5). Standard stringency levels were applied to ensure the relevance and reliability of the interactions. The resulting networks, showing enriched processes, were then visualized using Cytoscape software (version 3.7.2). We acknowledge the valuable contribution of the STRING database in our research and will duly reference it in the *Methods* section of our revised manuscript.

Revised text in line 638 of page 31: Protein–protein interactome information was collected from the STRING public database (version 11.5) to build a network showing enriched processes. A network model was built from tear DEPs and interactome data using Cytoscape software (version 3.7.2)⁵⁵.

Q28. Please comment on the multiplexing possibility and price of SNAFIA.

We appreciate the reviewer suggesting the need to discuss the multiplexing potential and price competitiveness of the SNAFIA test. We believe that such a discussion would add depth to the discussion of the utility and scalability of our immunodiagnostic platform for a broad range of diagnostic applications. Above all, our SNAFIA platform was designed with multiplexing potential to enable the simultaneous detection of multiple biomarkers in a single sample. Here, we propose two approaches for multiplexing. First, magnetic nanoparticles of different diameters (100 – 300 nm) can be used to guide the separation of nanoparticles bound to different biomarkers through magnetophoresis^{18, 19, 20}. This approach can be used with microfluidic chip systems. In this regard, we have stand-alone technologies and know-how in the synthesis and surface modification of magnetic nanoparticles and the design of channels in microfluidic chips for controllable magnetic separation^{18, 20}. Therefore, when applied, these technologies can separate magnetic nanoparticles of different diameters bound to different biomarkers into their respective channels in microfluidic chips.

In the second approach, multiplexing can be achieved by introducing multicolor FRET dye pairs encapsulated in polymer nanoprobcs. FRET dye pairs include multicolored organic dyes, quantum dots, and semiconductive polymeric nanoparticles. In particular, quantum dots are one of the inorganic nanoparticles widely used in FRET-based multiplexing designs due to their excellent fluorescence properties, such as broad absorption, high quantum yields, and narrow emission ranges depending on composition and size^{21, 22, 23}. We expect to be able to detect multiple biomarkers by separating biomarker-conjugated particles by size-driven magnetic separation and then analyzing the fluorescence signals derived from the polymer probe-magnetic nanoparticle complexes.

SNAFIA is an immunodiagnostic platform that integrates a magnetic separation system and fluorescent nanoprobcs, and due to the flexibility and scalability of this nanoparticle design, it has the potential to generate multiple separable optical signals targeting different biomarkers from a single sample. This multiplexing capability not only simplifies the diagnostic process but also provides test cost and time efficiencies.

Regarding the cost of SNAFIA, it is important to note that pricing can vary depending on several factors, including the number of biomarkers tested, the complexity of the assay, and the scale of production. However, one of our main goals was to develop a diagnostic tool that is affordable and easily accessible to users. As a result, based on 96 tests, the magnetic nanoparticles and polymer nanoprobcs cost \$130 each, and the wash buffer and lysis buffer cost \$5 each, bringing the total cost of the SNAFIA test to \$270 (approximately \$2.8 per test). Considering that commercially available immunoassays such as ELISA cost a total of \$900 for 96 tests (approximately \$9.4 per test), our test offers a relatively competitive price. As we continue to improve the performance of the SNAFIA platform

and apply it to mass production, we expect costs to decrease further, making it an affordable and practical diagnostic technique for widespread clinical use.

We believe that this multiplexing potential and cost-effectiveness will position the SNAFIA technology as a promising tool for future diagnostic applications in neurodegenerative diseases such as AD, which will be one of our crucial future research goals. We have added a section in the revised manuscript on the multiplexing feasibility and cost of the SNAFIA test and the future research directions, and we hope that this response sufficiently addresses the reviewer's comments.

New text in line 517 of page 28: SNAFIA is an immunodiagnostic platform that integrates a magnetic separation system and fluorescent nanoprobe, and due to the flexibility and scalability of this nanoparticle design, it has the potential to generate multiple separable optical signals targeting different biomarkers from a single sample. This multiplexing capability not only simplifies the diagnostic process but also provides test cost and time efficiencies.

New text in line 575 of page 29: The cost of the SNAFIA platform depends on factors like the number of biomarkers, assay complexity, and production scale. To enhance affordability and accessibility, we aimed for a cost-effective solution. For 96 tests, magnetic nanoparticles and polymer nanoprobe are \$130 each, while wash and lysis buffers are \$5 each, totaling \$270 (\$2.8 per test). In comparison, the cost per test is more than three times lower than traditional immunoassays such as ELISA. With ongoing platform enhancements and mass production, we anticipate further cost reductions, making SNAFIA a practical and economical diagnostic tool for widespread clinical use.

Q29. Please comment based on literature data the CAP1 protein. Was it discovered as a potential biomarker in other diseases, is it normally present in tears and other body fluids, etc.?

We thank the reviewer for the comments. CAP1 is integral to the remodeling of the actin cytoskeleton, and deficiency of this protein in the brain is associated with impaired neuronal development²⁴. Furthermore, CAP1 plays a pivotal role in the actin dynamics of neurons and the proper functioning of the growth cones²⁵. Given its fundamental role in neuronal differentiation and function, perturbations in CAP1 levels or function may contribute to the synaptic dysfunction that characterizes AD. A recent study by Zhong *et al.* highlighted differential expression of CAP1 in exosomes derived from the serum of AD patients²⁶. Although the exact role of CAP1 in AD remains unknown, the altered expression pattern of CAP1 suggests a potential involvement in the pathogenesis or progression of AD.

In the context of oncology, CAP1 is associated with tumor progression and metastasis across a range of cancer types²⁷. Additionally, CAP1 present on vascular and macrophage membranes influences the inflammatory responses of monocytes²⁸, potentially contributing to conditions such as coronary artery disease, immune disorders, metabolic disturbances, and pulmonary diseases.

Although there are a few reports of CAP1 being detected in body fluids such as serum^{26, 29} and

bronchoalveolar lavage fluid³⁰, there is little literature that reports explicitly the presence of CAP1 in tears or its potential as a biomarker for diverse diseases. Therefore, this study is the first to explore CAP1 as a potential tear biomarker for AD diagnosis, underscoring the novelty and importance of the findings.

Based on the above mention, we have supplemented the literature data with new references for the CAP1 protein in 'Discovery of human tear fluid biomarker by proteomic analysis' in the *Results and discussion* section to strengthen the rationale for CAP1 protein as a candidate biomarker for AD.

New reference in the *Results and discussion* section:

[35] Kakurina G. V., Kolegova E. S. & Kondakova I. V. Adenylyl Cyclase-Associated Protein 1: Structure, Regulation, and Participation in Cellular Processes. *Biochemistry (Mosc)* **83**, 45-53 (2018).

[36] Lee S. et al. Adenylyl cyclase-associated protein 1 is a receptor for human resistin and mediates inflammatory actions of human monocytes. *Cell Metab* **19**, 484-497 (2014).

[37] Schneider F. et al. Mutual functional dependence of cyclase-associated protein 1 (CAP1) and cofilin1 in neuronal actin dynamics and growth cone function. *Prog Neurobiol* **202**, 102050 (2021).

[39] Kakurina G. et al. A pilot study of the relative number of circulating tumor cells and leukocytes containing actin-binding proteins in head and neck cancer patients. *J Biomed Res* **37**, 213-224 (2022).

[40] Xie S. S., Hu F., Tan M., Duan Y. X., Song X. L. & Wang C. H. Relationship between expression of matrix metalloproteinase-9 and adenylyl cyclase-associated protein 1 in chronic obstructive pulmonary disease. *J Int Med Res* **42**, 1272-1284 (2014).

[41] Rust M. B. & Marcello E. Disease association of cyclase-associated protein (CAP): Lessons from gene-targeted mice and human genetic studies. *Eur J Cell Biol* **101**, 151207 (2022).

[46] Huang J. et al. Blood levels of MCP-1 modulate the genetic risks of Alzheimer's disease mediated by HLA-DRB1 and APOE for Alzheimer's disease. *Alzheimers Dement* **19**, 1925-1937 (2023).

[47] Sun R. & Xie C. Peripheral ApoE4 Leads to Cerebrovascular Dysfunction and Abeta Deposition in Alzheimer's Disease. *Neurosci Bull* **39**, 1330-1332 (2023).

Reviewer #2:

The fluorogenic immunoassay designed by the authors and the exhaustive controls carried out, do this MS interesting by itself. They demonstrate that is a useful technique to detect proteins in tears. Taking into account its sensibility, this technique can be useful for early diagnosis and monitoring of Alzheimer's disease from tear fluid or any other ophthalmic problem. However, to affirm that CAP-1 detected by this technique can be used as a biomarker for AD are necessary more additional data.

We thank the reviewers for taking the time to provide detailed and constructive feedback on our manuscript. We are pleased that the reviewer recognized that the SNAFIA test is useful for the early diagnosis and monitoring of Alzheimer's disease (AD) by detecting proteins in tear fluid. We hope the responses below adequately address the reviewer's points and satisfy the concerns raised.

Thus,

Q1. The mass spectrometry proteomics data have to be deposited in a public repository.

We thank the reviewer for this comment. We have deposited our mass spectrometry proteomics data in the PRIDE database, which is publicly accessible under the project accession number PXD042142. We have included this accession information in the *Data availability* section of the revised manuscript.

New text in line 797 of page 39: The mass spectrometry proteomics data have been deposited to the ProteomeXchange Consortium via the PRIDE repository with the dataset identifier (PXD042142).

Q2. Information with respect to the time to obtain the tear samples, tear volume and more important the presence of ophthalmic problems such as ophthalmic allergy, ophthalmic infection, ocular hypertension, Dry eye, and/or cataracts have to be published. AD and MCI used to have these problems.

We appreciate the reviewer's suggestions regarding the information about tear samples and the time of collection. We took careful steps to ensure the quality of the tear samples and their relevance to AD or mild cognitive impairment (MCI). First, we excluded patients with dry eye disease because it can affect the protein composition of tears. We also excluded patients with a history of ocular abnormalities in both eyes, including ocular surgery, injury, infection, allergies, inflammation (such as uveitis), glaucoma, and retinal diseases, as these conditions can cause alterations in tear composition independent of AD or MCI. In addition, we did not include patients using topical eye drops other than artificial tears, as these can alter the tear film and protein content, and we did not include contact lens wearers, as lens wear can influence the tear protein profiles. We also excluded people with systemic conditions such as autoimmune disorders, diabetes, or vascular disease that could affect tear composition. To maintain the integrity of our findings, we ensured that the study participants were free

of the aforementioned potential confounders. This information has been detailed in the *Methods* section of the revised manuscript to add clarity to the information about the quality of the tear samples.

New text in line 599 of page 30: Patients with dry eye disease were excluded, as it can influence the protein composition of tears. Those with any history of ocular abnormalities in either eye, including ocular surgery, injury, infection, allergy, inflammation (such as uveitis), glaucoma, or retinal diseases were also excluded, as these conditions can introduce variations in tear composition unrelated to AD or MCI. Patients were not included using any topical eyedrops other than artificial tears and contact lens wearers. Additionally, individuals with systemic diseases like autoimmune disorders, diabetes, and vascular diseases, which can have systemic effects were not part of our study.

Q3. ApoE4 genotype for human samples should be given

Thank you for raising this point. As detailed in our previous response to Reviewer 1's questions #7 and #8, the primary objective of this study was to validate the test efficacy of the SNAFIA platform in detecting protein biomarkers in human biofluids, particularly tears. To achieve this goal, we utilized APOE, a well-known protein associated with AD development in blood or cerebrospinal fluid (CSF), rather than a specific subtype such as APOE4. We want to clarify that this was a methodological choice to demonstrate the screening power of the SNAFIA test to broadly detect potential biomarkers in tears rather than to distinguish between specific APOE alleles.

Accordingly, in the revised manuscript, we have clarified the aims of the study and provided a more precise description of the specificity of the antibody used to detect APOE.

Revised text in line 274 of page 14: To evaluate the utility of the SNAFIA in human serum, Ab-MNPs and Ab-PNPs were coated with IgG antibodies that recognize all APOE proteins, not just the APOE4 subtype.

Q4. To ensure the quality of the samples, the presence of well-established tear markers such as lysozyme, lipocalin 1, serotransferrin or retinal dehydrogenase 1 should be published, This is important because slight epithelial damage caused by sampling is always taking place

The reviewer raised important points. We agree with the reviewer emphasizing the importance of mentioning the presence of several well-established tear markers in the tear proteomic profiling analysis because the delicate epithelium of the eye is frequently damaged by sampling procedures.

Indeed, while validating the quality of our samples for proteomic analysis, we checked for changes in the expression of tear proteins, including lysozyme, lipocalin-1 (LCP1), serotransferrin, and aldehyde dehydrogenase. The results confirmed the presence of these proteins in the participants' tears, but no significant fold change in protein expression was observed in the AD and MCI groups compared

to the healthy control (HC). Therefore, we excluded these proteins from the classification of differentially expressed proteins (DEPs) in this study. We also excluded these markers from the list of potential AD biomarkers suitable for validation of SNAFIA's test efficacy because they displayed inconsistent expression trends when comparing MCI/HC and AD/HC.

In addition, for LCP1, although it met the DEP criteria in terms of fold change and p -value in the MCI/HC vs. AD/HC comparison, the expression pattern remained relatively consistent between the two groups (1.57-fold and 1.56-fold change, respectively). Based on these flat trend results, we decided not to consider LCP1 as a potential biomarker for AD progression.

We are confident in the representativeness and high quality of the samples used in this study because we have extensive experience with human tear sampling and subsequent proteomic analysis, as shown in our previous publications^{1,2}, and we took great care to minimize potential epithelial damage to the ocular surface during the study procedures. Therefore, we have included a statement in the *Methods* section of the revised manuscript to clearly and transparently disclose the quality of the samples used in this study.

Q5. The authors talk about “the critical role of CAP1 pathogenesis AD”, but only a reference is given and related to its presence in tear samples.

We appreciate the reviewer for this comment. As mentioned in our response to question #29 of Reviewer 1, CAP1 plays a fundamental role in the remodeling of the actin cytoskeleton, which is crucial for neuronal development and function. Dysregulation of this protein could potentially lead to synaptic dysfunction, which is a hallmark of AD. A recent study by Zhong *et al.* highlighted the differential expression of CAP1 in exosomes derived from the serum of AD patients²⁶. Although the specific role of CAP1 in AD pathogenesis and progression is still under investigation, the markedly altered expression pattern suggests a potentially important role in the onset or progression of AD. CAP1 is also known to be involved in a variety of other diseases, including tumor progression in various cancer types and influencing inflammatory responses in conditions such as coronary artery disease and pulmonary diseases²⁷.

Although there are reports of CAP1 being detected in body fluids such as serum and bronchoalveolar lavage fluid, there have been no reports of this protein being detected in tears, so the possibility of CAP1 as a tear sample biomarker for AD is a novel aspect of this study. Therefore, in the revised manuscript starting in lines 187 page 10, we have discussed the value of CAP1 as a potential diagnostic biomarker for AD in more depth, which we hope will provide readers with a comprehensive understanding of its potential as a non-invasive and important diagnostic tool in AD and many other diseases.

Based on the reviewer's feedback, we have added literature related to the role of CAP1 in AD and the presence of CAP1 in various biofluids to strengthen the rationale for CAP1 protein as a candidate biomarker for AD.

New reference in the Results and discussion section:

[35] Kakurina G. V., Kolegova E. S. & Kondakova I. V. Adenylyl Cyclase-Associated Protein 1: Structure, Regulation, and Participation in Cellular Processes. *Biochemistry (Mosc)* **83**, 45-53 (2018).

[36] Lee S. et al. Adenylyl cyclase-associated protein 1 is a receptor for human resistin and mediates inflammatory actions of human monocytes. *Cell Metab* **19**, 484-497 (2014).

[37] Schneider F. et al. Mutual functional dependence of cyclase-associated protein 1 (CAP1) and cofilin1 in neuronal actin dynamics and growth cone function. *Prog Neurobiol* **202**, 102050 (2021).

[39] Kakurina G. et al. A pilot study of the relative number of circulating tumor cells and leukocytes containing actin-binding proteins in head and neck cancer patients. *J Biomed Res* **37**, 213-224 (2022).

[40] Xie S. S., Hu F., Tan M., Duan Y. X., Song X. L. & Wang C. H. Relationship between expression of matrix metalloproteinase-9 and adenylyl cyclase-associated protein 1 in chronic obstructive pulmonary disease. *J Int Med Res* **42**, 1272-1284 (2014).

[41] Rust M. B. & Marcello E. Disease association of cyclase-associated protein (CAP): Lessons from gene-targeted mice and human genetic studies. *Eur J Cell Biol* **101**, 151207 (2022).

[46] Huang J. et al. Blood levels of MCP-1 modulate the genetic risks of Alzheimer's disease mediated by HLA-DRB1 and APOE for Alzheimer's disease. *Alzheimers Dement* **19**, 1925-1937 (2023).

[47] Sun R. & Xie C. Peripheral ApoE4 Leads to Cerebrovascular Dysfunction and Abeta Deposition in Alzheimer's Disease. *Neurosci Bull* **39**, 1330-1332 (2023).

Q6. It is confusing to test the technique against ApoE (authors present many experimental controls) or AchEase and not to test them in human samples. If there are no changes, that is informative. With respect to ApoE could be interesting to test if, taking into account the genotype, a decrease or increase in ApoE is observed.

We thank the reviewer for the valuable suggestions. First, we would like to emphasize that our study focused on the analysis of human tear samples. Our proteomics analysis showed that the expression levels of APOE and ACHE proteins were insignificant to be used as valid biomarkers of tear fluid samples. For this reason, these proteins were designated as controls to demonstrate the detection ability of our detection platform for a wide range of proteins.

Unfortunately, due to the many challenges involved in obtaining human samples, we were unable to perform the tests requested by the reviewer, but we recognize the potential for confounding SNAFIA test results for APOE and ACHE, as highlighted by the reviewer. We have therefore moved all data related to APOE to the attached *Supplementary Information* document and added **Supplementary Fig. 15**, which is the objective of our study scenario: discovery and test validation of CAP1 as a potential

biomarker for the diagnosis of AD pathogenesis.

Q7. A table with all the proteins analyzed should be shown, at least in a supplementary table and with the name of them.

We thank the reviewer for pointing this out. As suggested by the reviewer, we have included **Table R1** in the *Supplementary Information* of the revised manuscript as detailed in our previous response to Reviewer 1's questions #23.

New Table R1:

No.	DEPs		
	Gene Symbol (MCI_AD_Common_DEP)	Fold change (MCI/HC)	Fold change (AD/HC)
1	RETN	2.34	1.58
2	CST2	2.22	1.86
3	SERPINA2	2.05	1.86
4	AGR2	2.02	1.64
5	BASP1	1.96	1.63
6	CES1	1.87	1.53
7	LRRC59	1.76	1.77
8	ALCAM	1.73	1.60
9	CAP1	1.72	1.86
10	GLUL	1.71	1.69
11	NAMPT	1.70	1.77
12	PDIA3	1.68	1.84
13	HMGB1	1.66	1.82
14	HIST1H1B	1.65	1.56
15	PSMB10	1.65	2.78
16	CAMP	1.65	1.69
17	SUB1	1.64	1.66
18	APOA4	1.63	1.65
19	VAPB	1.61	1.80
20	TOP1	1.61	1.68
21	HMGA1	1.60	1.74
22	FBL	1.60	2.16
23	AARS	1.60	1.67
24	KTN1	1.60	1.78
25	HSPB1	1.59	1.92
26	LMNB2	1.59	1.72
27	DNPEP	1.58	1.81
28	PSMA4	1.57	2.60
29	UTRN	1.57	1.57
30	ARGLU1	1.57	1.68
31	MARCKS	1.57	1.51
32	LCP1	1.57	1.56

33	TMA7	1.57	1.64
34	STK10	1.57	1.54
35	FAHD2A	1.57	1.50
36	ERP29	1.56	1.72
37	MANF	1.55	1.52
38	CCDC50	1.54	1.71
39	TROVE2	1.54	1.96
40	RALY	1.54	1.67
41	UBLCP1	1.54	2.09
42	TXNDC5	1.54	1.53
43	TAF15	1.53	1.66
44	BDH2	1.53	1.61
45	PRPSAP1	1.53	1.65
46	LAGE3	1.53	1.63
47	IGHV3-20	1.53	1.55
48	ILF3	1.53	1.72
49	SERPINC1	1.52	1.59
50	DSG2	1.52	1.61
51	TLN1	1.52	1.63
52	PTBP1	1.52	1.71
53	SNRPA	1.52	1.53
54	IGHG3	1.52	1.57
55	HSPA5	1.52	1.70
56	EEA1	1.51	1.57
57	HMGH4	1.51	1.66
58	STX4	1.51	1.98
59	CORO1A	1.51	1.72
60	SNRNP70	1.51	1.82
61	TCERG1	1.50	1.60
62	ACTB	1.50	1.59
63	PDAP1	1.50	1.50
64	CLINT1	1.50	1.63
65	PLA2G2A	0.67	0.54
66	LCN1	0.67	0.58
67	B2M	0.66	0.65
68	LACRT	0.63	0.52
69	PIP	0.55	0.57
70	PSCA	0.52	0.37
71	PRELP	0.49	0.48
72	SCGB1D1	0.48	0.43
73	SCGB2A1	0.46	0.43
74	IGKV1-9	0.39	0.38
75	SMR3B	0.36	0.49

Supplementary Table 2. Differentially expressed proteins (DEPs) in tear fluid of MCI and AD compared to HC. Sixty-four proteins exhibiting fold changes greater than 1.5 classified as upregulated (red) and eleven proteins showing a fold change less than 0.67 classified as downregulated (blue). Statistical analysis was performed for each protein using a *t*-test ($p < 0.01$).

Q8. Fig. 5B: Data should be corrected by protein content or at least by volume of tear. I understand that this correction has not been performed.

We thank the reviewer for the thoughtful advice. The experimental results presented in **Fig. 5** were obtained using a diluted tear sample, where 8 μL of the patient's tear sample was dispersed in 92 μL of artificial tear fluid (ATF) to keep the total volume of the tear samples at 100 μL . Here, the 8 μL of tear is a portion of the tear (approximately 10 μL) collected using the polyester wick.

New text in line 747 of page 36: For the SNAFIA test using clinical samples from the verification cohort, 8 μL of the human tear sample was dispersed in 92 μL of artificial tear fluid and prepared in advance.

Q9. Authors represent CAP-1 levels vs MMSE. Taking into account that they have also performed brain imaging, including 18F- florbetaben positron emission tomography scans could be interesting to do the same representations

We thank the reviewer for the insightful suggestions. As the reviewer suggests, correlating CAP1 levels in tears with pathophysiological amyloid deposition in the brain would provide a more holistic understanding of the role of CAP1 as a potential biomarker in AD. As elaborated in our response to Reviewer 1's comment #3, we are in the process of preparing a manuscript that delves into the specific interaction between CAP1 tear levels and brain amyloid or tau accumulation by PET imaging. We are confident that this follow-up study will further elucidate the intricate relationship between tear biomarkers and brain pathophysiology in AD.

Minor:

Q1. Fig4B, explain what grey circles means

Thank you for this careful observation. The inner and outer gray circles represent the relative protein expression (fold change) of each AD and MCI group compared to the HC group, with values close to 1.5 and -1.5 on the log₂ fold change scale shown in red and blue, respectively. We have added a detailed description of the gray circles in the revised figure caption accordingly.

New text in line 160 of page 9: The inner and outer gray circles represent the relative protein expression (fold change) of each AD and MCI group compared to the HC group, with values close to 1.5 and -1.5 on the log₂ fold change scale shown in red and blue, respectively.

Q2. Could be expressed data Fig 5B in absolute terms? You can do a calibration curve with CAP-1 and then express the concentration of CAP1 in your sample.

As commented by the reviewer, we converted the data in **Fig. 5b** to concentrations based on the

calibration curve (4-parameter logistic curve), shown in **Fig. 4c**, and presented the results in the table below. As shown in **Table R2**, some human tear samples in **Fig. 5b** were outside the dynamic range of the calibration curve (10^{-1} – 10^3 fM), so the calculated CAP1 concentration could not be obtained. Similarly, it is worth noting that results with negative normalized fluorescence signals were also not converted to CAP1 concentrations. Therefore, we would like to draw the reviewer’s attention to the difficulty of converting all results in **Fig. 5b** to CAP1 concentrations using the calibration curve. Testing more clinical samples to expand the detection range and attempt to quantify this SNAFIA test will be one of the important aspects of our future work.

New Table R2:

(1) HC group:

1	2	3	4	5	6	7	8	9	10	11	12	13	14
1.04 E+00	1.01 E+00	n.d.	1.03 E+00	n.d.	1.06 E+00	1.14 E+00	n.d.	1.02 E+00	1.04 E+00	1.01 E+00	1.01 E+00	1.00 E+00	1.07 E+00
1.04 E+00	1.01 E+00	n.d.	1.02 E+00	n.d.	1.07 E+00	1.09 E+00	n.d.	1.02 E+00	1.04 E+00	1.02 E+00	1.01 E+00	1.01 E+00	1.07 E+00
1.05 E+00	1.01 E+00	n.d.	1.00 E+00	n.d.	1.05 E+00	1.09 E+00	n.d.	1.03 E+00	1.05 E+00	n.d.	1.01 E+00	1.01 E+00	1.06 E+00

(2) MCI group:

15	16	17	18	19	20	21	22	23	24	25	26	27	28	29
1.3E +00	n.d.	n.d.	2.2E +04	6.2E +00	n.d.	n.d.	1.2E +00	N.D.	n.d.	2.8E +00	n.d.	1.8E +00	n.d.	n.d.
1.2E +00	n.d.	n.d.	7.4E +04	5.4E +00	n.d.	n.d.	1.2E +00	1.0E +00	n.d.	2.5E +00	n.d.	1.8E +00	n.d.	n.d.
1.2E +00	n.d.	n.d.	1.1E +05	6.9E +00	n.d.	n.d.	1.2E +00	1.0E +00	n.d.	2.3E +00	n.d.	2.1E +00	n.d.	n.d.

(3) AD group:

30	31	32	33	34	35	36	37	38	39
n.d.	2.08 E+00	n.d.	n.d.	n.d.	n.d.	n.d.	n.d.	1.04 E+00	n.d.
n.d.	1.92 E+00	n.d.	n.d.	n.d.	n.d.	n.d.	n.d.	1.04 E+00	n.d.
2.77 E+04	1.86 E+00	n.d.	n.d.	2.77 E+04	n.d.	n.d.	n.d.	1.05 E+00	n.d.

Table R2. Calculated CAP1 concentrations in human tear samples from HC (patient ID: 1–14), MCI (patient ID: 15–29), and AD groups (patient ID: 30–39) used in the verification cohort. The CAP1 concentration was obtained by calculating the normalized fluorescence intensity (FI) values from SNAFIA based on a calibration curve with known CAP1 concentrations in ATF. n.d., not determined.

Fig. R5b:

Fig. R5b. SNAFIA test results showing the calculated CAP1 concentrations in human tear samples from HC (left), MCI (middle), and AD (right) using a calibration curve with known concentration of CAP1 in ATF. All measurements were performed in triplicate, and data represent mean \pm s.d. n.d., not determined.

Q3. Fig 5b (MCI group): Pink box is explained in line 480, but should be explained in the figure legend).

We thank the reviewer for this comment. Based on the reviewer's comment, we have inserted a sentence in the figure caption indicating that the sample marked with a pink box (patient ID: 16) is a patient who has progressed from MCI to definite AD in the last 2 years.

New text in line 489 of page 25: The pink box labeled patient (patient ID: 16) showed disease progression from MCI to definite AD over two years.

Reviewer #4:

The authors report a new FRET sensing platform for the rapid detection of AD biomarkers in tears. It is based on magnetic nanoparticles functionalized with the detection antibodies. The article starts from the synthesis and characterization of the NPs, continues with the sensing principles and finishes with the tests performed on real samples.

The sensing principle and the functionalization procedure are not novel. But the biological target is original and of utmost importance for the early diagnosis of AD. The analytical performances are remarkable. The diagnosis could be based on the identification of proteins in tears with a very good sensitivity. How does it compare with digital ELISA such as the Simoa platform? The obtained results are convincing; the manuscript is clear and well-written. Even if the sensing strategy is not completely novel, considering the quality of the bioanalytical results, the importance and the potential medical and societal impact of the reported immunoassays, I support the publication of this manuscript in Nature Communications.

We sincerely thank the reviewer for taking the time to provide a positive evaluation of our work. In response to the reviewer's inquiry regarding the comparison to digital ELISA, we provide our response below.

Microwell magnetic bead-based digital enzyme-linked immunosorbent assay (ELISA) platforms, exemplified by technologies such as Quanterix's single-molecule arrays (Simoa®), are similar to our SNAFIA platform in that they utilize magnetic beads for immunocomplex formation and rely on a fluorescence signal-driven detection mechanism²⁵. Despite these similarities, we would like to point out that our SNAFIA is distinct from the Simoa technologies. First, digital assays like Simoa require disks with specially designed microwell patterns for sample compartmentalization, expensive automation equipment, and multiple procedures such as repeated washing and incubation, adding complexity, hands-on time, and cost. In contrast, **SNAFIA is an in-solution mix-and-read assay that requires only microtubes, a magnet, and laboratory microplates to conduct the entire workflow, making the process from sample preparation to signal detection relatively fast, less complex, and cost-effective.** Second, a unique aspect of SNAFIA lies in its enzyme-free detection system. While both traditional and digital ELISAs apply an enzyme-substrate reaction to amplify the signal, **the SNAFIA platform leverages the signal amplification mechanism of polymeric nanoprobe that utilize Förster resonance energy transfer (FRET) without the intervention of enzymes.** This FRET-based fluorescence detection approach overcomes artifacts due to enzyme kinetics, substrate specificity, and temperature dependence, and eliminates the need for separate development of chromogenic or chemiluminescent signals, leading to reduced assay time and reproducible results. Therefore, these differences collectively characterize the unique attributes of the SNAFIA approach over conventional and digital ELISA techniques.

Once again, we appreciate the reviewer for the opportunity to present the novelty of our work and hope the raised concerns are sufficiently addressed.

Minor point:

-Fig. 2e : it should be « Immunolabeled... » instead of « Immunolabel... »

We appreciate this comment. We have corrected the word 'Immunolabel' to 'Immunolabeled' in **Fig. 2e (Fig. 3f** in the revised manuscript).

Revised Fig. 2:

Fig. 3 Characterization of the synthesized Ab-MNPs and Ab-PNPs. (a) Schematic of the immunocomplex constructed by Ab-MNPs and Ab-PNPs in the presence of the target protein (CAP1) in tear fluid. (b) Schematic of magnetic separation of Ab-MNPs using a neodymium magnet in the SNAFIA test. (c) Representative transmission electron microscopy (TEM) image of Ab-MNPs dispersed

in deionized water (DW). The scale bar represents 200 nm. (d) Powder X-ray diffraction (XRD) patterns of the MNPs. A diffraction pattern of the reported Fe₃O₄ (JCPDS. 01-017-4918) is also shown. (e) Schematic representation of Ab-MNPs labeled with immunogold (Ab-AuNPs) to better visualize the primary capture antibody bound to the MNPs. (f) Representative TEM image of Ab-MNPs labeled with immunogold (Ab-AuNPs, approximately 10 nm in diameter). Red arrows indicate the localized AuNPs on the surface of Ab-MNPs. The scale bars indicate 100 nm and 50 nm (inset), respectively. (g) Magnetization curves of MNPs and Ab-MNPs obtained by vibrating sample magnetometer (VSM) at 25 °C. (h) Schematic of FRET signal changes of Ab-PNPs induced by treatment with TX-100 surfactant as lysis buffer. (i) TEM image of Ab-PNPs negatively stained with 3% (w/v) phosphotungstic acid solution (pH 6.81). The scale bar represents 100 nm. (j) Emission fluorescence spectra of Ab-PNPs before and after treatment with lysis buffer (excitation at 475 nm). Data represent mean ± s.d. for three independent experiments. Schematics were created with BioRender.com.

References

1. Jung J. H. et al. Proteomic analysis of human lacrimal and tear fluid in dry eye disease. *Sci Rep* **7**, 13363 (2017).
2. Ji Y. W. et al. Changes in Human Tear Proteome Following Topical Treatment of Dry Eye Disease: Cyclosporine A Versus Diquafosol Tetrasodium. *Invest Ophthalmol Vis Sci* **60**, 5035-5044 (2019).
3. Posa A., Brauer L., Schicht M., Garreis F., Beileke S. & Paulsen F. Schirmer strip vs. capillary tube method: non-invasive methods of obtaining proteins from tear fluid. *Ann Anat* **195**, 137-142 (2013).
4. Willcox M. D. P. et al. TFOS DEWS II Tear Film Report. *Ocul Surf* **15**, 366-403 (2017).
5. Zhou L. et al. Identification of tear fluid biomarkers in dry eye syndrome using iTRAQ quantitative proteomics. *J Proteome Res* **8**, 4889-4905 (2009).
6. Broadhurst D. I. & Kell D. B. Statistical strategies for avoiding false discoveries in metabolomics and related experiments. *Metabolomics* **2**, 171-196 (2006).
7. Halim A., Nilsson J., Ruetschi U., Hesse C. & Larson G. Human urinary glycoproteomics; attachment site specific analysis of N- and O-linked glycosylations by CID and ECD. *Mol Cell Proteomics* **11**, M111 013649 (2012).
8. Skates S. J. et al. Statistical design for biospecimen cohort size in proteomics-based biomarker discovery and verification studies. *J Proteome Res* **12**, 5383-5394 (2013).
9. Thomas S., Hao L., Ricke W. A. & Li L. Biomarker discovery in mass spectrometry-based urinary proteomics. *Proteomics Clin Appl* **10**, 358-370 (2016).
10. Serrano-Pozo A., Das S. & Hyman B. T. APOE and Alzheimer's disease: advances in genetics, pathophysiology, and therapeutic approaches. *Lancet Neurol* **20**, 68-80 (2021).
11. Raulin A. C., Doss S. V., Trottier Z. A., Ikezu T. C., Bu G. & Liu C. C. ApoE in Alzheimer's disease: pathophysiology and therapeutic strategies. *Mol Neurodegener* **17**, 72 (2022).
12. Huang J. et al. Blood levels of MCP-1 modulate the genetic risks of Alzheimer's disease mediated by HLA-DRB1 and APOE for Alzheimer's disease. *Alzheimers Dement* **19**, 1925-1937 (2023).
13. Sun R. & Xie C. Peripheral ApoE4 Leads to Cerebrovascular Dysfunction and Abeta Deposition in Alzheimer's Disease. *Neurosci Bull* **39**, 1330-1332 (2023).
14. Gouilly D. et al. Beyond the amyloid cascade: An update of Alzheimer's disease pathophysiology. *Rev Neurol (Paris)*, (2023).
15. Hansen S. B. & Wang H. The shared role of cholesterol in neuronal and peripheral inflammation. *Pharmacol Ther* **249**, 108486 (2023).
16. Na H. et al. Peripheral apolipoprotein E proteins and their binding to LRP1 antagonize Alzheimer's disease pathogenesis in the brain during peripheral chronic inflammation. *Neurobiol Aging* **127**, 54-69 (2023).
17. Kallo G. et al. Changes in the Chemical Barrier Composition of Tears in Alzheimer's Disease Reveal Potential Tear Diagnostic Biomarkers. *PLoS One* **11**, e0158000 (2016).
18. Kang B. et al. Immunomagnetic microfluidic integrated system for potency-based multiple separation of heterogeneous stem cells with high throughput capabilities. *Biosens Bioelectron* **194**, 113576 (2021).
19. Lee H. et al. Rapid Visible Detection of African Swine Fever Virus Using Hybridization Chain Reaction-Sensitized Magnetic Nanoclusters and Affinity Chromatography. *Small*, e2207117 (2023).

20. Mun B. et al. An immuno-magnetophoresis-based microfluidic chip to isolate and detect HER2-Positive cancer-derived exosomes via multiple separation. *Biosens Bioelectron* **239**, 115592 (2023).
21. Park C. et al. Kinetic stability modulation of polymeric nanoparticles for enhanced detection of influenza virus via penetration of viral fusion peptides. *J Mater Chem B* **9**, 9658-9669 (2021).
22. Park G. et al. Cell-mimetic biosensors to detect avian influenza virus via viral fusion. *Biosens Bioelectron* **212**, 114407 (2022).
23. Park C. et al. Membrane Rigidity-Tunable Fusogenic Nanosensor for High Throughput Detection of Fusion-Competent Influenza A Virus. *Advanced Functional Materials* **33**, (2023).
24. Rust M. B. & Marcello E. Disease association of cyclase-associated protein (CAP): Lessons from gene-targeted mice and human genetic studies. *Eur J Cell Biol* **101**, 151207 (2022).
25. Schneider F. et al. Mutual functional dependence of cyclase-associated protein 1 (CAP1) and cofilin1 in neuronal actin dynamics and growth cone function. *Prog Neurobiol* **202**, 102050 (2021).
26. Zhong J. et al. Discovery of Novel Markers for Identifying Cognitive Decline Using Neuron-Derived Exosomes. *Front Aging Neurosci* **13**, 696944 (2021).
27. Kakurina G. V., Kolegova E. S. & Kondakova I. V. Adenylyl Cyclase-Associated Protein 1: Structure, Regulation, and Participation in Cellular Processes. *Biochemistry (Mosc)* **83**, 45-53 (2018).
28. Lee S. et al. Adenylyl cyclase-associated protein 1 is a receptor for human resistin and mediates inflammatory actions of human monocytes. *Cell Metab* **19**, 484-497 (2014).
29. Kakurina G. et al. A pilot study of the relative number of circulating tumor cells and leukocytes containing actin-binding proteins in head and neck cancer patients. *J Biomed Res* **37**, 213-224 (2022).
30. Xie S. S., Hu F., Tan M., Duan Y. X., Song X. L. & Wang C. H. Relationship between expression of matrix metalloproteinase-9 and adenylyl cyclase-associated protein 1 in chronic obstructive pulmonary disease. *J Int Med Res* **42**, 1272-1284 (2014).

REVIEWERS' COMMENTS

Reviewer #1 (Remarks to the Author):

The authors uploaded the proteomics data to PRIDE, addressed most of my questions, and made a substantial effort to reorganize the manuscript, however, the description of nanoparticle generation in the results part is still very extensive. I think moving the figures from this part to the supplementary material greatly improved the manuscript and in this form, it is more clear what the scientific message is, however, it could be further improved.

I know how hard is to collect tear samples from patients with AD. Understanding the explanations of the authors, still I cannot fully accept them. I think that for such an important method, which hopefully could be used even in clinical settings for diagnosis of AD and maybe ocular diseases, demonstration with statistical probes that the relatively low number of patients (and here if I understand well, only the validation cohorts should be taken into account – figure 5) is enough is indispensable.

Otherwise, the manuscript now is well written, and if the results of the sample size determination are acceptable, I support its publication.

Reviewer #2 (Remarks to the Author):

The authors have substantially and adequately addressed comments/criticisms made of the original MS.

Reviewer #4 (Remarks to the Author):

The authors have significantly improved the manuscript in this revision. They have provided a reasonable response to each of the previous referee concerns and they have made suitable revisions where required. Again, the authors developed a bioanalytical strategy for the rapid detection of AD biomarkers in tears with magnetic nanoparticles. The results are very impressive and will lead to further developments in the field of immunoassays, diagnostics, nanomaterials, etc. The CL duration time is very

large and this sets the bar very high for non-enzymatic systems. The work is well presented and of high quality. I recommend its publication in the present form.

Response to Reviewers' Comments

We are sincerely grateful for the reviewers' thoughtful and constructive feedback on our paper. The insightful comments have significantly enriched the quality and depth of our research, contributing to its overall improvement. Your dedication to the peer-review process is greatly appreciated, and we look forward to addressing your suggestions to make our paper even more robust and valuable to the academic community. Thank you for your time and expertise.

Reviewer #1:

Q1. The authors uploaded the proteomics data to PRIDE, addressed most of my questions, and made a substantial effort to reorganize the manuscript, however, the description of nanoparticle generation in the results part is still very extensive. I think moving the figures from this part to the supplementary material greatly improved the manuscript and in this form, it is more clear what the scientific message is, however, it could be further improved.

We extend our gratitude to the reviewer for recognizing our efforts. Numerous of the observations you provided have propelled our research toward a more sophisticated trajectory.

As mentioned in our initial rebuttal letter, our main purpose is to prove the efficacy of the SNAFIA test, which incorporates nanoparticles. Therefore, introducing the design of SNAFIA and the results of particle characterization is one of the sections that should be mainly covered in this study. Nonetheless, we also consider the reviewer's comments as a valuable guide for enhancing our study. Therefore, we have rearranged the results to the attached *Supplementary Information* document and added **Supplementary Notes 1 and 2**, which are the descriptions of nanoparticle generation results.

Revised Supplementary Fig. 1:

Supplementary Fig. 1 Characterization of the synthesized antibody-immobilized magnetic nanoparticles (Ab-MNPs). (a) Schematic of antibody immobilization process. (b) Transmission electron microscope (TEM) images of the **magnetic nanoparticles (MNPs)** series dispersed in deionized water (DW). The scale bar represents 200 nm (inset, 20 nm). (c) Hydrodynamic size distribution of the MNPs series dispersed in DW at a 0.1 mg/mL concentration determined by dynamic light scattering (DLS). (d) Hydrodynamic diameters of the MNPs series. (e) **Powder X-ray diffraction patterns of the MNPs. A diffraction pattern of the reported Fe₃O₄ (JCPDS. 01-017-4918) is also shown.** Data represent mean ± s.d. of three independent experiments ($n = 3$). Statistical analysis was performed by multiple comparisons of Dunn's analysis of variance tests (ns, non-significant). **Schematics were created with BioRender.com.**

Revised Fig. 2:

Fig. 3 Characterization of the synthesized Ab-MNPs and Ab-PNPs. **(a)** Schematic of the immunocomplex constructed by Ab-MNPs and Ab-PNPs in the presence of the target protein (CAP1) in tear fluid. **(b)** Schematic representation of Ab-MNPs labeled with immunogold (Ab-AuNPs) for better visualization of the primary capture antibody bound to the MNPs. **(c)** Representative transmission electron microscopy (TEM) image of Ab-MNPs labeled with immunogold (Ab-AuNPs, approximately 10 nm in diameter). Red arrows indicate the localized AuNPs on the surface of Ab-MNPs. The scale bars indicate 100 nm and 50 nm (inset), respectively. **(d)** Magnetization curves of MNPs and Ab-MNPs obtained by vibrating sample magnetometer (VSM) at 25 °C. **(e)** Schematic of Förster resonance energy transfer (FRET) signal changes of Ab-PNPs induced by treatment with TX-100 surfactant as lysis buffer. **(f)** TEM image of Ab-PNPs negatively stained with 3% (w/v) phosphotungstic acid solution (pH 6.81). The scale bar represents 100 nm. **(g)** Emission fluorescence spectra of Ab-PNPs before and after treatment with lysis buffer (excitation at 475 nm). Data represent mean \pm s.d. for three independent experiments. Schematics were created with BioRender.com.

Q2. I know how hard is to collect tear samples from patients with AD. Understanding the explanations of the authors, still I cannot fully accept them. I think that for such an important method, which hopefully could be used even in clinical settings for diagnosis of AD and maybe ocular diseases, demonstration with statistical probes that the relatively low number of patients (and here if I understand well, only the validation cohorts should be taken into account – figure 5) is enough is indispensable.

Otherwise, the manuscript now is well written, and if the results of the sample size determination are acceptable, I support its publication.

We appreciate the reviewer's concerns and recognize the importance of ensuring the statistical significance of our sample size, especially in the verification cohort presented in **Figure 5**.

One of the fundamental prerequisites for a successful quantitative study is to ensure the analysis possesses adequate statistical power to yield meaningful results¹. Power is defined as $1-\beta$, where β represents the false-negative rate. This refers to the probability that the result is a false negative rate, indicating the likelihood of a Type II error^{2, 3, 4}.

Figure R1 shows a graphical representation of the required expression difference as fold change versus the number of biological replicates in each group⁵. These graphs were formulated for 25, 50, 75, and 100% variation at both technical and biological levels. The results were generated using the `pwr.2p.test()` function in the free statistical software package R ([www. r-project.org](http://www.r-project.org)), with a power set at 0.8 and a confidence level of 0.05. The effect size was calculated as the expression difference divided by the total variation^{5, 6}.

From the differentially expressed proteins (DEPs) listed in **Table R1**, the fold change values of the CAP1 protein were quantified at 1.72 and 1.86 in the tear fluid of MCI and AD patients compared to HC, respectively. Applying our results to the power analysis data, the minimum sample size required for a power analysis would be 15 for MCI and 10 for AD when compared to HC. In the verification cohort, the SNAFIA was performed on HC ($n = 14$), MCI ($n = 15$), and AD ($n = 10$) subjects. Given the adequate sample size selected, the results obtained underscore the efficacy of SNAFIA as a diagnostic tool.

It's also worth noting the challenges associated with collecting tear samples from AD patients, which the reviewer acknowledged. In conclusion, this study serves as a proof-of-concept, necessitating larger-scale clinical investigations to comprehensively validate the clinical potential of our approach. In the manuscript, we will ensure that the manuscript explicitly details the methodology employed for sample size determination and emphasizes the importance of more extensive clinical studies.

New text in line 528 of page 26: While this study serves as a proof-of-concept, it underscores the need for more extensive clinical investigations to comprehensively validate the clinical potential of this approach. If further larger-scale clinical studies confirm these findings, SNAFIA is simple, fast, and provides high diagnostic accuracy for a variety of protein markers using tear fluid, making it a promising tool for early diagnosis of AD.

New text in line 597 of page 29: Applying our proteomics analysis to the power analysis data⁵²,

⁵³, the minimum sample size required for a power analysis would be 15 for MCI and 10 for AD when compared to HC.

New Figure R1:

Fig. R1 Power analysis for 25, 50, 75, and 100% variation. The function `pwr.2p.test()` in the statistical software package R was used to generate the data. The power was set to 0.8, with confidence at 0.05.

New Table R1:

No.	DEPs		
	Gene Symbol (MCI_AD_Common_DEP)	Fold change (MCI/HC)	Fold change (AD/HC)
1	RETN	2.34	1.58
2	CST2	2.22	1.86
3	SERPINA2	2.05	1.86
4	AGR2	2.02	1.64
5	BASP1	1.96	1.63
6	CES1	1.87	1.53
7	LRRC59	1.76	1.77
8	ALCAM	1.73	1.60
9	CAP1	1.72	1.86
10	GLUL	1.71	1.69
11	NAMPT	1.70	1.77
12	PDIA3	1.68	1.84
13	HMGB1	1.66	1.82
14	HIST1H1B	1.65	1.56
15	PSMB10	1.65	2.78
16	CAMP	1.65	1.69
17	SUB1	1.64	1.66

18	APOA4	1.63	1.65
19	VAPB	1.61	1.80
20	TOP1	1.61	1.68
21	HMGA1	1.60	1.74
22	FBL	1.60	2.16
23	AARS	1.60	1.67
24	KTN1	1.60	1.78
25	HSPB1	1.59	1.92
26	LMNB2	1.59	1.72
27	DNPEP	1.58	1.81
28	PSMA4	1.57	2.60
29	UTRN	1.57	1.57
30	ARGLU1	1.57	1.68
31	MARCKS	1.57	1.51
32	LCP1	1.57	1.56
33	TMA7	1.57	1.64
34	STK10	1.57	1.54
35	FAHD2A	1.57	1.50
36	ERP29	1.56	1.72
37	MANF	1.55	1.52
38	CCDC50	1.54	1.71
39	TROVE2	1.54	1.96
40	RALY	1.54	1.67
41	UBLCP1	1.54	2.09
42	TXNDC5	1.54	1.53
43	TAF15	1.53	1.66
44	BDH2	1.53	1.61
45	PRPSAP1	1.53	1.65
46	LAGE3	1.53	1.63
47	IGHV3-20	1.53	1.55
48	ILF3	1.53	1.72
49	SERPINC1	1.52	1.59
50	DSG2	1.52	1.61
51	TLN1	1.52	1.63
52	PTBP1	1.52	1.71
53	SNRPA	1.52	1.53
54	IGHG3	1.52	1.57
55	HSPA5	1.52	1.70
56	EEA1	1.51	1.57
57	HMGN4	1.51	1.66
58	STX4	1.51	1.98
59	CORO1A	1.51	1.72
60	SNRNP70	1.51	1.82
61	TCERG1	1.50	1.60
62	ACTB	1.50	1.59
63	PDAP1	1.50	1.50
64	CLINT1	1.50	1.63
65	PLA2G2A	0.67	0.54

66	LCN1	0.67	0.58
67	B2M	0.66	0.65
68	LACRT	0.63	0.52
69	PIP	0.55	0.57
70	PSCA	0.52	0.37
71	PRELP	0.49	0.48
72	SCGB1D1	0.48	0.43
73	SCGB2A1	0.46	0.43
74	IGKV1-9	0.39	0.38
75	SMR3B	0.36	0.49

Supplementary Table 2. Differentially expressed proteins (DEPs) in tear fluid of MCI and AD compared to HC. Sixty-four proteins exhibiting fold changes greater than 1.5 classified as upregulated (red) and eleven proteins showing a fold change less than 0.67 classified as downregulated (blue). Statistical analysis was performed for each protein using a *t*-test ($p < 0.01$).

Reviewer #2:

The authors have substantially and adequately addressed comments/criticisms made of the original MS.

Reviewer #4:

The authors have significantly improved the manuscript in this revision. They have provided a reasonable response to each of the previous referee concerns and they have made suitable revisions where required. Again, the authors developed a bioanalytical strategy for the rapid detection of AD biomarkers in tears with magnetic nanoparticles. The results are very impressive and will lead to further developments in the field of immunoassays, diagnostics, nanomaterials, etc. The CL duration time is very large and this sets the bar very high for non-enzymatic systems. The work is well-presented and of high quality. I recommend its publication in the present form.

We extend our gratitude to the remaining two reviewers for their valuable contributions to our research. Their insightful comments have played a pivotal role in enhancing the clarity and emphasis of our strategic approach. We wish to reiterate our appreciation for the dedicated time and effort invested by all of our esteemed reviewers.

References

1. Nakayasu E. S. et al. Tutorial: best practices and considerations for mass-spectrometry-based protein biomarker discovery and validation. *Nat. Protoc.* **16**, 3737-3760 (2021).
2. Karp N. A. et al. Maximising sensitivity for detecting changes in protein expression: experimental design using minimal CyDyes. *Proteomics* **5**, 3105-3115 (2005).
3. Molloy M. P. et al. Overcoming technical variation and biological variation in quantitative proteomics. *Proteomics* **3**, 1912-1919 (2003).
4. Roy S. M. et al. Differential expression profiling of serum proteins and metabolites for biomarker discovery. *Int. J. Mass Spectrom.* **238**, 163-171 (2004).
5. Levin Y. The role of statistical power analysis in quantitative proteomics. *Proteomics* **11**, 2565-2567 (2011).
6. Karp N. A. & Lilley K. S. Design and analysis issues in quantitative proteomics studies. *Proteomics* **7 Suppl 1**, 42-50 (2007).